# Arms races between selfish genetic elements and their host defence in termites

Bitao Qiu [1] ✉, Daniel Elsner [1] & Judith Korb [1,2] ✉

Arms races between parasites and hosts are key drivers of evolution. Selfishly replicating transposable elements (TEs) are thought to follow similar dynamics, but strong evidence is missing. We test this in termites, social insects in which TEs have been linked to ageing. Sequencing genomes and profiling DNA methylation across the termite phylogeny reveal corresponding phylogenetic signals in TEs and TE methylation, indicative of co-evolution. TE methylation reduces TE success, as both TE abundance and spreading efficiency decrease with increasing methylation. TEs also become less harmful with TE age: evolutionarily older TEs spread less, insert less into exons, and erode into short remnants. Correspondingly, defence through methylation is strongest against young TEs. Yet, as in typical host–parasite arms races, some TEs persist, implying resistance or recurrent invasions. Our results reveal arms races between TEs and DNA methylation, positioning TEs as drivers of genome evolution similar to symbionts in organismic evolution.

Conflict can lead to antagonistic evolutionary arms races. When organisms continuously reduce their partners' fitness, they impose selection for defence mechanisms that can result in adaptations and counter-adaptations between co-evolving partners[1]. This is best known for prey/predator and host/parasite interactions. One of the best-studied examples is bats and moths. Bats use echolocation for moth-prey detection and moths evolved the ability to detect bat calls or confuse bats with click calls to evade capture[2,3]. We predict similar arms races at the organism level for genetic elements that behave selfishly, for instance, by replicating independently within a genome, which harms host fitness[4–6]. Yet, evidence for full-cycle evolutionary arms races is scarce, especially for the most abundant selfish genetic elements, transposable elements (TEs)[6,7].

TEs, often called 'jumping genes', can either (i) make a copy of themselves via an RNA intermediate that inserts at a new genomic position ('copy and paste' mechanism; retrotransposons, class I elements) or (ii) change their position within the host genome by excising themselves from one location and inserting into another ('cut' and 'paste' mechanism; DNA Transposons, class II elements)[8]. Furthermore, TEs have been classified into superfamilies and families based on sequence similarity[9]. Although TEs can sometimes provide genetic material for evolutionary innovation[10–13], their activity is commonly

costly to the individual bearing them ('the host'). TEs can compromise genome functioning when jumping into genes or regulatory regions or cause costs through ectopic recombination[14–16]. Hence, TEs are commonly regarded as genomic parasites[5,6,17]. As expected for harming parasites, counter-mechanisms have evolved that protect the host against TEs by silencing TEs transcriptionally or post-transcriptionally[6,18–21]. The most important mechanisms are the piwi-interacting RNA (piRNA) pathway, which silences active TEs[22], and chromatin marks and DNA methylation, two conserved epigenetic mechanisms that repress TEs[6,23–27].

Although being a controversial topic[28], in many organisms somatic TE activity has been associated with ageing and age-related diseases, including termites[29–32]. Termites are social insects with genomes typically composed of about 50% TEs[33,34]. Regardless of caste, senescent (old-aged) termite individuals are characterized by high TE activity that present a large potential threat. However, the long-lived queens and kings are better protected against TE activity than the short-lived workers[31,35]. This combination of high TE content and TE-associated caste-specific ageing makes termites a compelling system for studying the co-evolution between TEs and their hosts. This differs compared to social Hymenoptera like ants and bees, which harbour much fewer TEs in their genomes[36]. In addition, DNA methylation,

[1]Evolutionary Biology & Ecology, University of Freiburg, Freiburg, Germany. [2]Research Institute for the Environment and Livelihoods, Charles Darwin University, Casuarina Campus, Darwin, Australia. ✉e-mail: bitao.qiu.88@gmail.com; judith.korb@biologie.uni-freiburg.de

which can constitutively silence TEs[6,26], is especially prevalent in termites in contrast to many other insects[34,37,38].

We test for evolutionary arms races between TEs and TE defence via DNA methylation in termites. We employ long-read sequencing, which overcomes biases from bisulfide short-read sequencing in profiling DNA methylation in repetitive regions like TEs[39–41]. We simultaneously sequence the genomes and measured DNA methylation levels from the same individuals of seven termite species across the termite phylogeny, along with six replicate population samples of a focal species. By analysing the co-evolution pattern of TEs and TE DNA methylation in termites, we trace the evolutionary history of TEs and TE defence. We show that the harm of TEs to the host generally declines with TE age, while host defence through DNA methylation is strongest against young TEs. Our results reveal strong evidence for evolutionary arms races between TE and TE defence in termites.

## Results

### TEs and TE methylation patterns have strong evolutionary signals in termites

We used PacBio HiFi long-read systems to sequence high-molecular-weight DNA extracted from eight single individual samples of seven termite species that cover the full spectrum of termite social complexity. These species comprised *Mastotermes darwiniensis*, *Zootermopsis nevadensis*, *Cryptotermes secundus*, *Reticulitermes grassei*, *Trinervitermes geminatus*, *Macrotermes bellicosus* and *Odontotermes* sp.2 (Fig. 1a). For *M. bellicosus*, we sequenced both a queen and a soldier sample as species replication and to compare a potential caste effect on TE abundance and DNA methylation (see Supplementary Note 1). For each individual, we generated 17 to 40 Gb HiFi reads, with mean read length from 5028 to 16,689 bp and estimated sequencing depth from 16 to 66x (Supplementary Table 1). We assembled our termite HiFi reads and obtained eight genome assemblies with sizes ranging from 576 Mb in *Z. nevadensis* to 1.4 Gb in *Odontotermes* sp.2 (Fig. 1a) ("Method"). Our genome assemblies had a genome completeness (coverage of the total BUSCO gene content) between 98.2% and 100% (Fig. 1a) (for further genome quality data, see also Supplementary Table 1), achieving high quality genome assemblies from single individuals.

To examine the distribution of TEs across the termite phylogeny, we built a non-redundant repeat library and annotated the repeat content (including TEs) in termites, including our eight termite genome assemblies and the published genome assembly of a representative of their woodroach sister taxon, *Cryptocercus punctulatus*[42]. Repeat content in the termite genomes ranged from 40.5% in *Z. nevadensis* to 61.3% in *M. darwiniensis* (Fig. 1a). Termite genome size correlated significantly with both the total size and the genomic proportion of repetitive content (Pearson's $r = 0.99$ and 0.94, respectively, $n_{species} = 7$, both $p < 0.001$) (Supplementary Fig. 1). This result supports the hypothesis that genome size variation in termites is driven by TE expansion and contraction[33], as in many other organisms[43,44].

Focusing on TE superfamilies, principal component analysis (PCA) showed that TE abundance (i.e., the copy numbers of different TE superfamilies in the genome; a measure of the replication success of TEs, comprising cumulatively active and non-active TEs) largely reflects termite phylogeny[45,46] (Fig. 1b). Principle component (PC) 1 separated the Geoisoptera (Termitidae and *R. grassei*) from the other termites, while PC2 separated *Z. nevadensis* from *C. secundus* and *M. darwiniensis*. We found a similar pattern with a clustering analysis (Fig. 1c). Among the top TE superfamilies that contribute to the two PCs, many had lineage/species specific distributions (Supplementary Data 1). For example, Mutator-like elements (MULE) and Maverick, two DNA transposon superfamilies, were only highly abundant in the Geoisoptera (Supplementary Data 1). By contrast, hAT DNA transposons and Alu short interspersed nuclear elements (SINE) were restricted to *Z. nevadensis* (Fig. 1c) (Supplementary Data 1). This phylogenetic

TE signal indicates lineage/species-specific TE evolution. This result contradicts the evolutionary model that TE evolution is largely driven by random drift effects[47] but it is in line with an evolutionary arms race scenario.

To test whether we can detect a corresponding TE defence signal, we quantified TE DNA methylation. To this end, we used the kinetic information from the PacBio HiFi reads to generate the methylation (5mC) probabilities for CpG sites in our termite genome assemblies ("Method" and see Supplementary Note 2 for the validation of CpG methylation quantification). The genome-wide percentage of methylated CpG sites was not associated with termite phylogeny (Fig. 1a). However, CpG methylation levels (i.e. the percentages of methylated CpGs normalised by the whole genome background; hereafter TE methylation for simplicity) of TE superfamilies reflected the phylogeny of termites, showing a similar pattern as TE abundance. In a PCA, PC1 separated the Geoisoptera from *C. secundus* and PC2 separated Geoisoptera from *M. darwiniensis* and *Z. nevadensis* (Fig. 1d). Clustering analysis revealed a similar pattern with genomes from Geoisoptera clustering together in the dendrogram (Fig. 1e). In addition, the dendrogram based on TE methylation was largely consistent with that based on TE abundance (permutation test: $n_{permutation} = 1000$, $p < 0.01$) (Supplementary Fig. 2), as predicted by the evolutionary arms race hypothesis. These results show lineage/species-specific evolution of TE methylation but not of methylation in general, indicating that TE methylation is heritable, under natural selection and associated with TE abundance.

To gain further insights into potential drivers of these phylogenetic TE and TE methylation signals, we zoomed into individual TE superfamilies. Some TE superfamilies had consistent methylation patterns across all termite species (Supplementary Data 2). For example, LINE-2 and PiggyBac had high methylation levels across all genomes, while LINE-1 TEs were hypo-methylated (Supplementary Data 2). By contrast, other TE superfamilies revealed lineage/species-specific methylation patterns that were positively or negatively associated with their abundance in some parts of the phylogeny. For example, Pao LTR retrotransposons showed high abundance and high methylation levels in the Macrotermitinae, but low abundance and low methylation levels in *Z. nevadensis* and *M. darwiniensis*. In contrast, MULE and Maverick elements, which are both abundant in Geoisoptera, displayed low methylation levels in all species except *Z. nevadensis* and/or *M. darwiniensis* (Fig. 1e). This lineage/species-specific pattern can be explained by the evolutionary invasion history of the TE families (see Supplementary Note 3). Because methylation levels vary widely among TE families within each superfamily (Supplementary Fig. 3), finer-scale analyses at the TE family level are required to test for potential arms race dynamics.

### TE abundance indicates TE and TE methylation arms race dynamics in termites

To study the arms races at the TE family level, we estimated the evolutionary age of TE families (hereafter, TE age) based on their lineage-specificity across the termite phylogeny ("Method"). We focused on *M. bellicosus* because our genome samples included closely related species (Fig. 1a), which allow us to identify species/lineage-specific TE families. Among the 2144 classified TE families in *M. bellicosus*, 33% (704 families) can be traced back more than 120 Ma to the last common ancestor of *R. grassei* and *C. secundus* (Icoisoptera; hereafter, ancient TE families), while only 4% (85 families) were *M. bellicosus* specific (hereafter, young TEs) (Fig. 2a). The average TE sequence length significantly decreased with TE age (Spearman's $r = −0.23$, $n_{family} = 2144$, $p < 1e-5$) (Fig. 2b). TE length was over six times longer for young, *M. bellicosus*-specific TE families than for ancient TE families (1734 bp vs. 271 bp). This indicates that ancient TEs eroded or lost parts of their sequence, which could make them non-functional over time.

If there are evolutionary arms races between TEs and TE methylation, we predict young, species-specific TEs to be hyper-methylated

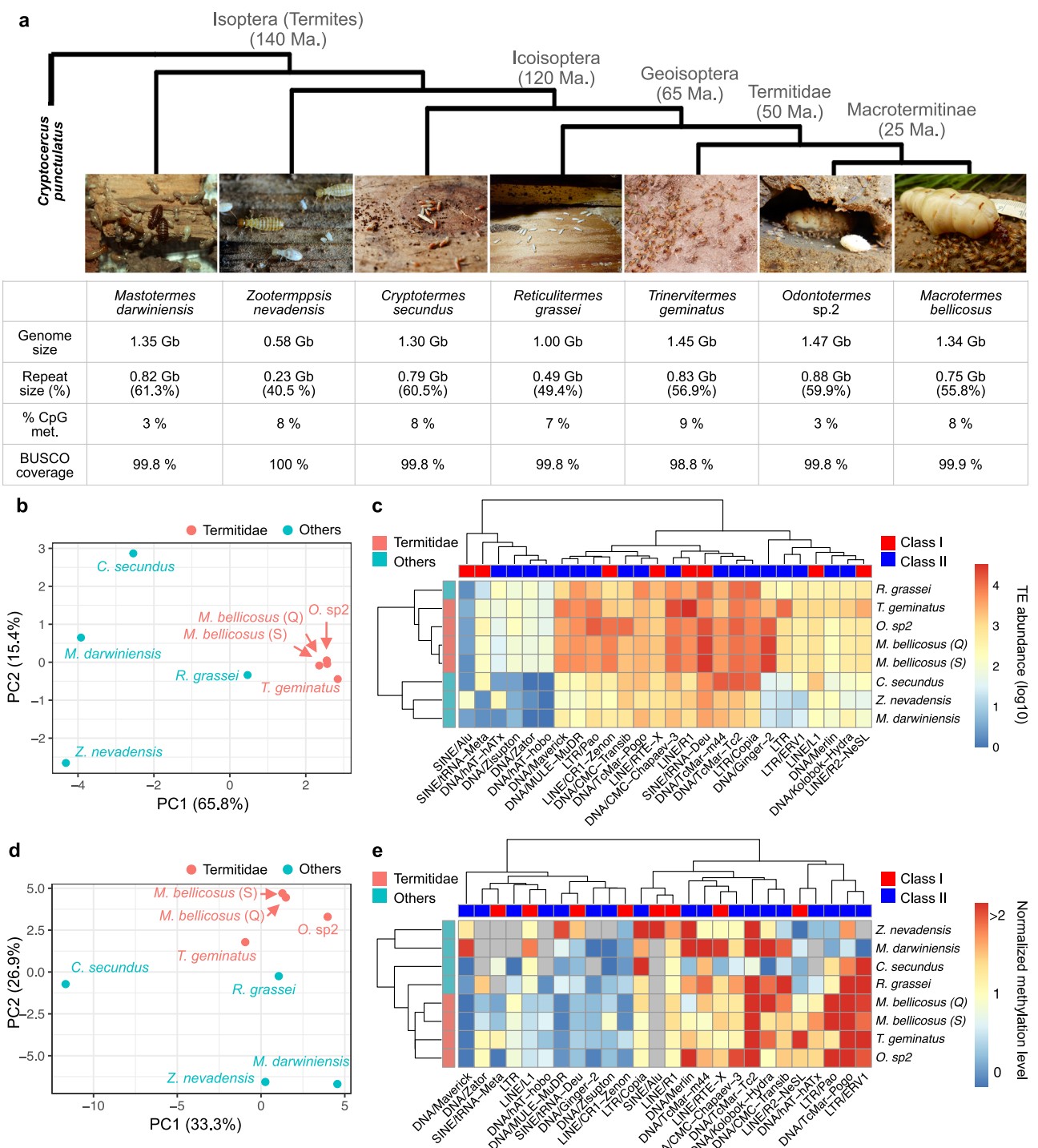

**Fig. 1 | TEs and TE methylation patterns have strong evolutionary signals in termites. a** Cladogram and overall genome statistics (genome size, repeat content, genome wide methylation level and BUSCO gene content completeness) of the seven studied termite species. The phylogenetic tree of termites is based on[45,46]. Termite photos were taken and provided by Judith Korb. **b** PCA of TE superfamily abundances (n = 71) in the termite genomes, coloured according to species taxonomy. **c** Heatmap of TE superfamily abundances for the top 26 TE superfamilies showing the highest variation in abundance or CpG methylation across species. Abundances are shown on a log₁₀ scale. **d** PCA of TE superfamily CpG methylation levels (n = 69) in the termite genomes, coloured according to termite taxonomy. **e** Heatmap of normalized CpG methylation levels for the same TE superfamilies as in panel **c**. Normalised methylation levels range from 0 (devoid of methylated CpG sites), 1 (equal to the genomic background), to > 2 (over two folds higher than the genomic background). The normalised methylation level was calculated as the ratio of the percentage of methylated CpG sites in a TE superfamily to that in the genome. TE superfamilies are coloured in grey when methylation information were unavailable due low abundance., For *M. bellicosus*, both a queen (Q) and a soldier (S) genome were included. Source data are provided as a Source Data file.

while ancient TE families are under-methylated. This is because young, newly invaded TEs are more active and harmful to the host[17,48]. By contrast, ancient TE families could be non-active due to erosion (Fig. 2b) or might even have been evolutionary co-opted and perform host functions[13,49]. To test this, we examined the effect of TE age on TE family methylation level in the *M. bellicosus* genome. Supporting an evolutionary arms race scenario, both variables negatively correlated (Spearman's $r = -0.21$, $n_{family} = 2144$, $p < 1e\text{-}5$) (Fig. 2c). Particularly,

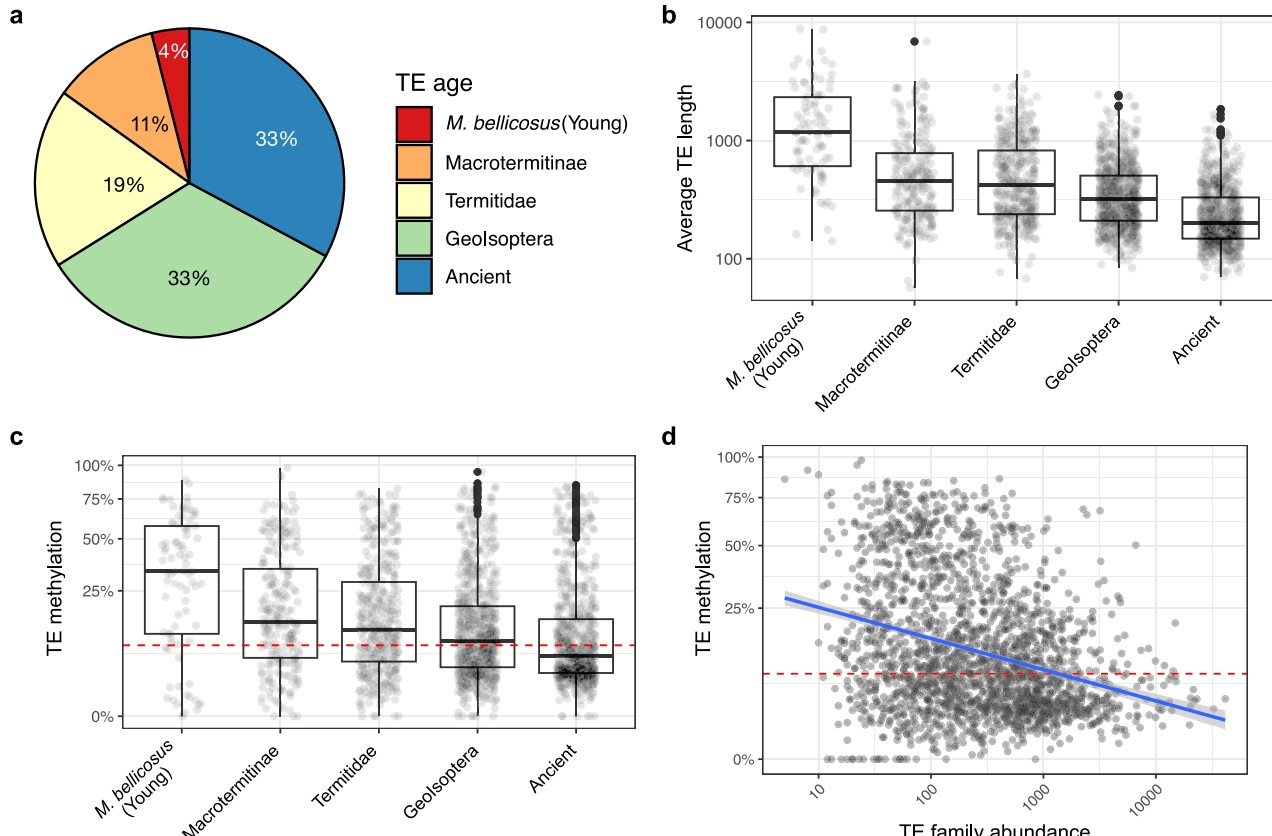

**Fig. 2 | Age dynamics of TE families and TE methylation in *M. bellicosus*. a** Pie-chart of the age distribution of TE families ($n = 2144$) in *M. bellicosus*. TE family ages were estimated based on the distribution of TE families across the termite phylogeny. 'Ancient' means TE families older than the last common ancestor of Neoisoptera and *C. secundus*. **b** Boxplots of average sequence length for TE families from different TE age groups. The average sequence length was calculated as the total sequence length of a TE family divided by its copy number. **c** Boxplots of the percentages of methylated CpG sites for TE families from different TE age groups. **d** The association between the percentage of TE methylation and TE family abundance ($n_{family} = 2144$). A generalized regression model was fitted (blue line), with a 95% confidence band. We found a similar pattern in other termite genomes (Supplementary Fig. 3). For **b** and **c**, boxplots show the median (centre line), the 25th and 75th percentiles (box boundaries), and whiskers extend to 1.5× the interquartile range (IQR). Points beyond the whiskers are shown as outliers. For **b, c** and **d**, each point represents one TE family. The number of TE families for the TE-age groups are: $n_{M. bellicosus} = 85$, $n_{Macrotermitinae} = 239$, $n_{Termitidae} = 403$, $n_{Geoisoptera} = 713$, $n_{ancient} = 704$. The red dashed lines in **c** and **d** represent the background genomic methylation level (8%) of *M. bellicosus*. Source data are provided as a Source Data file.

methylation levels of young *M. bellicosus*-specific TEs were significantly higher than the genomic background (median methylation level = 4.17, two-sided Wilcoxon rank-sum test: $V = 3368$, $n_{family} = 85$, $p < 1e\text{-}5$), while those of ancient TE families were similar (median methylation level = 0.73; two-sided Wilcoxon rank-sum test: $V = 128223$, $n_{family} = 704$, $p = 0.44$) (see Supplementary Fig. 4 for a similar pattern in other termite species).

We also stratified TE families into recently expanded and more diverged families based on their mean within-family sequence divergence, quantified by Kimura's distance (KD), an indicator of the time since TE invasion/expansion in a species[36,50,51] ("Method"). Recently expanded TE families generally contain highly similar TE copies with low mean KD values, whereas more-diverged families, which are likely non-active, show greater sequence divergence due to the accumulation of mutations, resulting in high mean KD values. Interestingly, while TE methylation levels generally declined with TE age, more-diverged TE families (KD > 20) had higher methylation levels than recently expanded TE families (KD < 20) across all TE age groups, except for ancient TE families that are likely eroded (Supplementary Figs. 5 and 6). This result is consistent with the expectation that DNA methylation suppresses TE proliferation and that methylation in ancient non-active families, which no longer pose a threat to the host, is no longer maintained by natural selection.

Furthermore, if TE methylation functions as a means to silence TEs, TE abundance should negatively correlate with TE methylation. As expected, TE abundance declined significantly with TE methylation (Spearman's $r = -0.29$, $n_{family} = 2144$, $p < 1e\text{-}5$) (Fig. 2d). This applied also after controlling for TE age, with a significant interaction between TE methylation and TE age (ANCOVA: $F_{(1, 2134)}$ TE methylation = 241.06, $p$ TE methylation < 1e-5; $F_{(4, 2134)}$ TE age = 5.60, $p$ TE age < 0.001; $F_{(4, 2134)}$ TE methylation x TE age = 2.87, $p$ TE methylation x TE age = 0.022), showing that the TE methylation effect is most pronounced in younger TEs (Supplementary Fig. 7). We also found similar negative correlations between TE methylation and TE abundance in other termite genomes, stressing the generality of the results (Supplementary Fig. 8).

## TE activity reveals TE and TE methylation arms race dynamics in *M. bellicosus*

To measure the impact of TEs in the genome, we determined the number of TE-associated structure variants (SVs). TE-associated SVs indicate potential harm to the host because SVs are measures of mutation. In addition, as SVs are quantified within a population of a species, they also hint at currently active TEs (see also Supplementary Note 4) and their numbers correlate with the TEs replication success, a measure of TE fitness. To this end, we sequenced individual genomes from six *M. bellicosus* colonies to identify SVs and infer TE activities

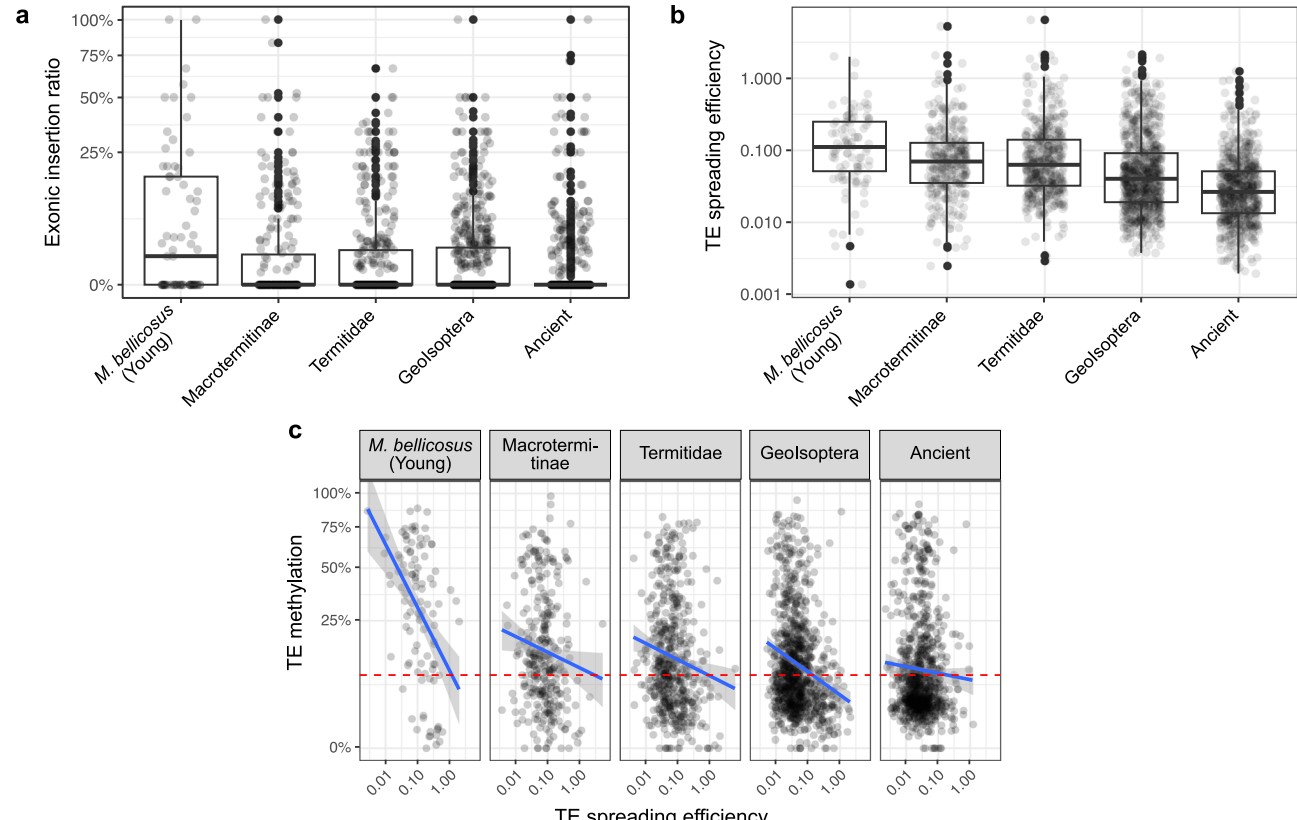

**Fig. 3 | TE activity reveals TE and TE methylation arms race dynamics in *M. bellicosus*. a** Boxplots of exonic insertion ratios of TE-associated SVs (structural variants) across different TE ages. The exonic insertion ratio was calculated as the percentage of exonic insertions to the total number of insertions per family. **b** Boxplots of the TE spreading efficiency across different TE ages. **c** The association between TE spreading efficiencies and TE methylation levels, plotted separately for each TE age group. Each age group was fitted with a generalized regression model (blue line) and a 95% confidence band. The red dashed line represents the

background genomic methylation level (8%) of *M. bellicosus*. For **a** and **b**, boxplots show the median (centre line), the 25th and 75th percentiles (box boundaries), and whiskers extend to 1.5× the interquartile range (IQR). Points beyond the whiskers are shown as outliers. For **a–c**, each point represents one TE family. The number of TE families for the TE-age groups are: $n_{M. bellicosus} = 71$, $n_{Macrotermitinae} = 239$, $n_{Termitidae} = 374$, $n_{Geoisoptera} = 572$, $n_{ancient} = 463$. For **b** and **c**, TE spreading efficiency was calculated as the number of TE-associated SVs normalised by TE family abundance. Source data are provided as a Source Data file.

("Method"). We found 267,001 SVs among the *M. bellicosus* samples and, by annotating the SVs with our termite repeat library, identified 172,157 (64%) TE-associated SVs ("Method").

We detected a striking observed abundance bias for the locations of TE-associated SVs that implies selection against harmful exonic insertions. Among the TE-associated SVs, only 1.5% (2249) were located in exons, while the rest were located in introns (55.2%) or intergenic (43.3%) regions. This observed abundance bias persisted after controlling for the sequence length of introns and exons, with TE-associated SVs 2.8 times less likely to be observed in exonic than in intronic regions (two-proportion Z-test: $\chi^2 (1) = 5304.8$, $p < 1e-5$, effect size = $-2.21 \times 10^{-5}$, 95% C.I. = $[-2.25 \times 10^{-5}, -2.18 \times 10^{-5}]$). Furthermore, comparing the exonic insertion ratio (i.e., the number of exonic to total insertions) across TE families revealed a significant decline with increasing TE family abundance (Spearman's $r = -0.28$, $n_{family} = 1719$, $p < 1e-5$) (Supplementary Fig. 9), indicating that exonic insertions are more strongly selected against and hence less successful. In addition, the exonic insertion ratio declined with TE family age (Spearman's $r = -0.1$, $n_{family} = 1719$, $p < 1e-5$) (Fig. 3a). In particular, the exonic insertion ratio for ancient TE families was almost four times lower than that for young, *M. bellicosus*-specific TE families (0.026 vs. 0.106; two-sided Wilcoxon rank-sum test: $W = 21487$, $n_{ancient\ family} = 463$, $n_{young\ family} = 71$, $p < 1e-5$), suggesting that ancient TEs are less harmful to the host.

On the opposite side, TE family success was associated with a young age. The proportion of highly active TE families (copy number

of TE-associated SVs > 3) was significantly higher in young, *M. bellicosus*-specific TE families than ancient TE families (71 out of 103 vs. 493 out of 845 TE families; 69.0% vs. 58.3%; Fisher's exact test: odds ratio = 1.58, $p = 0.043$, 95% C.I. = [1.00, 2.54]), as expected when young TEs are more likely to mobilize[52]. In addition, TE family spreading efficiency, measured as the ratio between TE-associated SVs and TE abundance (a measure of TE success) ("Method"), was negatively correlated with TE age (Spearman's $r = -0.34$, $n_{family} = 2687$, $p < 1e-5$) (Fig. 3b). This holds for DNA transposons or retrotransposons (Supplementary Fig. 10). Young, *M. bellicosus*-specific TE families were 3.2 times more successful in spreading than ancient TE families. Thus, young TE families are more likely to be active and are more efficient in spreading than older ones.

When TE DNA methylation functions as a defence mechanism that can suppress TEs, we predict a negative correlation between TE methylation and the spreading efficiency of TE families. As expected, over all TE age groups, TE families with high methylation had a lower TE spreading efficiency (Spearman's $r = -0.14$, $n_{family} = 2687$, $p < 5e-5$). Similar to the pattern with TE abundance (Fig. 2d), the effect of TE methylation on TE spreading efficiency applied even after controlling for the effect of TE age with a significant interaction effect between TE methylation and TE age (ANCOVA: $F(1, 2677)_{TE\ methylation} = 10.97$, $p_{TE\ methylation} < 0.001$; $F(4, 2677)_{TE\ age} = 21.97$, $p_{TE\ age} < 1e-5$; $F(4, 2134)_{TE\ methylation\ x\ TE\ age} = 2.45$, $p_{TE\ methylation\ x\ TE\ age} = 0.044$). The correlation coefficient between TE spreading efficiency and TE methylation was positively associated with TE age (Spearman's $r = 0.7$, $n_{age\ group} = 5$,

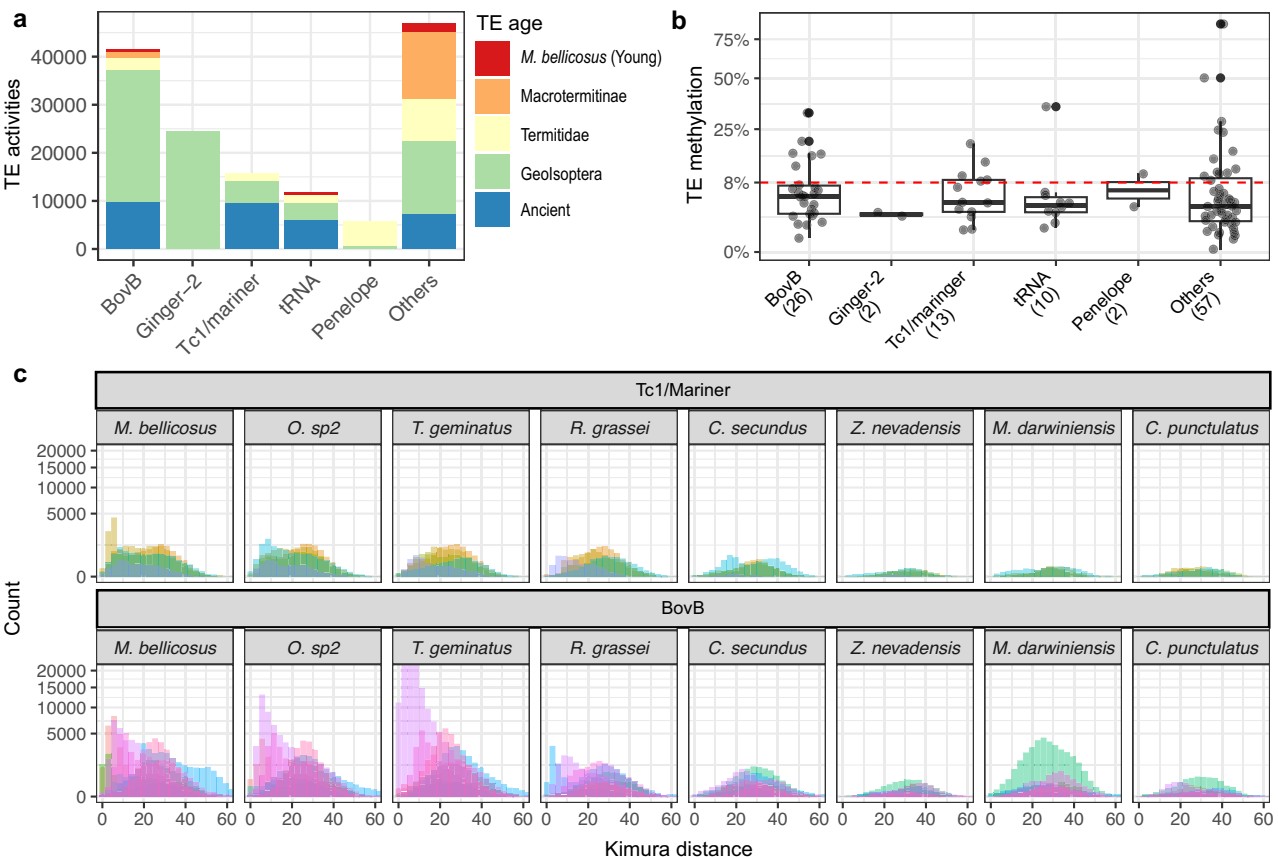

**Fig. 4 | Top active TE families can have ancient origins. a** Activity levels of the top 100 active TE families in *M. bellicosus*. TE families are grouped by superfamilies and coloured by TE age, ranging from young, *M. bellicosus*-specific (red) to ancient (blue) TE families. TE activity levels are quantified by the copy number of TE-associated SVs in a *M. bellicosus* population (*N* = 6 genomes). The top five active TE superfamilies are shown separately; the remaining are grouped as Others. **b** The percentages of methylated CpG sites for the top 100 active TE families, grouped by superfamilies. Only the top 5 most active TE superfamilies are shown, with the number of TE families within each superfamily provided in parentheses. The red dotted line represents the genomic methylation level (8%) of *M. bellicosus*. Boxplots show the median (centre line), the 25th and 75th percentiles (box boundaries), and whiskers extend to 1.5× the interquartile range (IQR). Points beyond the whiskers are shown as outliers. **c** Histograms of Kimura distances (KD) for the top active Tc1/Mariner (upper panel, *n* = 5) and BovB (lower panel, *n* = 8) ancient TE families, plotted separately for each species. Each colour represents one TE family. A low KD value indicates recent TE expansion/invasion and a high KD value suggests TE remnants that have been inactive for a long time. Source data are provided as a Source Data file.

*p* = 0.23), increasing from −0.46 in young, *M. bellicosus*-specific TE families to −0.04 in ancient TE families (Spearman correlation tests: *p* < 1e-5 and *p* = 0.23, respectively) (Fig. 3d). This implies that the selection pressure on the host to defend itself against TEs declines with TE age, in line with ancient TEs being less harmful, as indicated by their lower exonic abundance, reduced spreading efficiency, and sequence erosion.

## Top active TE families can have ancient origins

To analyse the origin of the top active TE families, we identified the top 100 TE families with the highest number of TE-associated SVs in the *M. bellicosus* genome. We found that BovB (LINE) (*n* $_{family}$ = 26) and Tc1/mariner (DNA transposon) (*n* $_{family}$ = 13) were among the top three active TE superfamilies (Fig. 4a), consistent with previous findings that BovB and Mariner are highly active in *M. bellicosus*[31]. In line with their high activity, their mean methylation levels were lower than that of the genomic background (Two-sided Student's *t*-tests: $p_{BovB}$ = 0.01, $p_{Tc1/mariner}$ = 0.05) (Fig. 4b). Interestingly, thirteen (33%) of these 39 top-active BovB and Tc1/mariner TE families were ancient TEs (Fig. 4a), suggesting top-active TEs can have ancient origins.

To trace the evolutionary histories of these ancient TE families, we further inspected the distribution of their within-family sequence divergence (KD values among TE copies within the same family). While mean KD values reflect the overall divergence of a TE family, the distribution of KD values within a family reveals the composition of its members. Consistent with their high activities in *M. bellicosus*, KD frequency distributions for the majority (9 out of 13) of these TE families had a mode with KD values lower than 10, indicative of recently expanded TE copies, and these modes were largely absent in the non-Termitidae genomes (Fig. 4c). This suggests that they were recent expansions/invasions in *M. bellicosus*, despite being ancient TEs. Seven of these nine TE families also had an additional second mode in *M. bellicosus* and a single mode in many other termite species, all of which had KD values higher than 20 (Fig. 4c and Supplementary Fig. 11), reflecting high sequence divergence probably due to many TE remnants (decayed, non-active TE copies) and ancient invasions. Our results imply that these seven TE families have copies of ancient origin and copies recently expanded in *M. bellicosus*.

## TE defence genes are under strong selection in the termite genomes

If TEs play a major role as parasites of termite genomes, we predict selection on TE defence genes. These include genes involved in DNA methylation and retrotransposon silencing, particularly those from the piRNA pathway that mediate DNA methylation to silence TEs[6,18,19].

Thus, we examined whether TE defence genes are under positive selection in our termite genomes, using the woodroach as outgroup[53,54] ("Method"). Forty-nine TE defence genes were present in

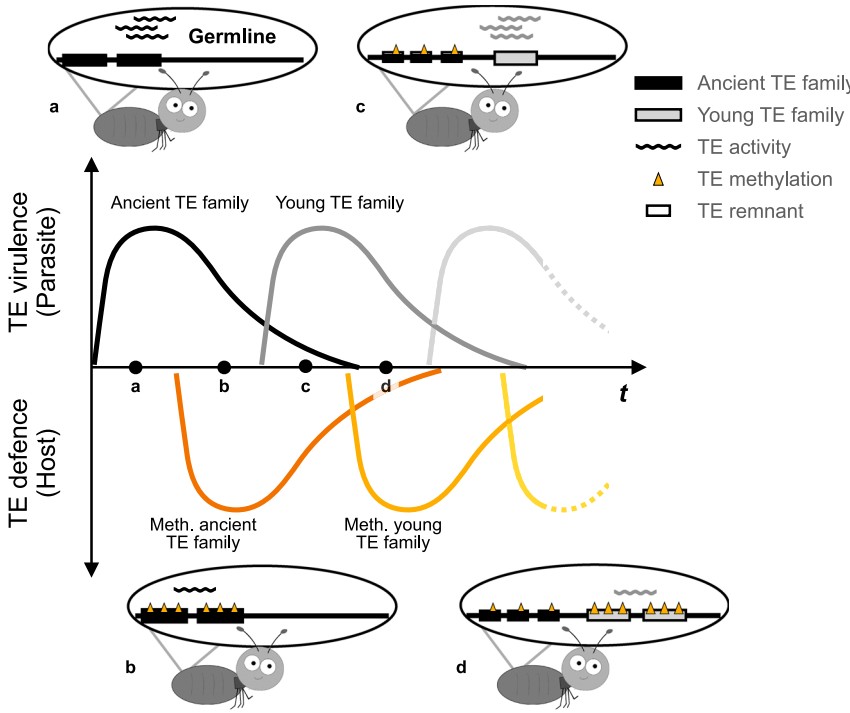

**Fig. 5 | A schematic diagram of the arms race between TEs and TE defence. a** TEs can behave as genomic parasites, whose unchecked proliferation can cause significant harm to their hosts. **b** In response, natural selection favours hosts that evolve molecular defence systems, such as DNA methylation, to suppress these genomic parasites and limit their spread. **c** Over evolutionary time, once-rampant ancient TE families gradually lose their virulence, by reduced spreading efficiency, fewer exonic insertions, and degeneration into TE remnants. As their threat wanes, so does the selective pressure on the host to maintain costly defences, leading to reduced TE methylation. Yet young TE families emerge, launching fresh invasions into the host genome. **d** This ever-renewing arms race selects hosts to once again ramp up their defences, reinforcing methylation-based suppression against new TE families. Termite illustration is adapted from[90].

both the woodroach and termites (Supplementary Data 3). Among these genes, 15 (31%) exhibited a signature of positive selection in the termites, which was significantly higher than expected based on the whole genome background (7%) (two-sided binomial test: $p < 1e\text{-}5$, 95% C.I. = [0.19, 0.46]). Ten (32%) out of the 31 piRNA pathway genes were under positive selection. For example, both aubergine (*Aub*) and *qin*, which are differentially expressed between young and old *M. bellicosus* workers[31], were under positive selection in the lineage leading to the Macrotermitinae. Several piRNA pathway genes have also been repeatedly selected in multiple termite lineages/species (Supplementary Fig. 12), indicating their important roles in the evolutionary arms races with TEs.

## Discussion

Our results revealed evolutionary arms races between TEs and TE methylation in termites (Fig. 5). Such arms races typically involve: (i) Parasites that harm a host, (ii) host defences against the parasite, (iii) co-evolution of parasite virulence and host defence, and (iv) evolutionary outcomes ranging from parasites being permanently 'shut down' and becoming commensals or even mutualists, to recurrent arms races[1,14,55,56]. We found support for all of these components.

### TEs as parasites
As former studies have also shown[33,34], TEs account for around 50% of termite genome (except *Z. nevadensis*) with genome size correlating with TE content (Supplementary Fig. 1 and Supplementary Note 5). This reflects their accumulation over time and their huge importance for their host. Their harm has been shown in ageing studies that associate ageing and senescence with TE activity and the extraordinarily long lifespan of the queen and king with TE defensive mechanisms[31,57].

### TE methylation as host defence against TEs
As predicted, when TE methylation suppresses TEs, our study showed that both TE abundance (i.e., active and formerly active TEs) and, more importantly, TE spreading efficiency (i.e., the success of currently active TEs) declined with increasing TE methylation in termites (Figs. 2d and 3b). Thus, our study also supports the emerging evidence that DNA methylation, although less common in some insect orders[58–60], can play an important role in insects and their defence against TE[23,60]. Importantly, as TE spreading efficiency can be seen as a direct measure of TE fitness, our result demonstrates that TE methylation impedes TE success. Thus, TE methylation is a host defence mechanism that ultimately benefits the host. Hence, our study establishes a direct link between DNA methylation and TE success.

### Co-evolution of virulence and host defence
We found clear evolutionary age effects, indicative of co-evolution between TEs and termite defence via TE methylation (i.e. they reciprocally affect each other through processes of natural selection). On average, TE virulence (harm to the host) declined with the evolutionary age of TE families, and accordingly, selection on host defence weakened as well.

Evidence for a decline of TE virulence comes from the following three results. First, the spreading efficiency of TEs - and hence their potential harm - declined with TE age (Fig. 3b). Because TE methylation negatively affected spreading efficiency (Fig. 3c), the age-related decline in TE spreading efficiency is likely, at least in part, a consequence of host defence. Second, although some ancient TEs remain active, most are probably inactive remnants, as indicated by their substantially reduced sequence length (Fig. 2b), low spreading efficiency (Fig. 3b) and high sequence divergence (Supplementary Fig. 5). Third, exonic TE insertions, which are generally more harmful than

insertions in other regions[61], were less common in ancient than in younger TEs (Fig. 3a). This systematic difference in exonic insertion bias likely reflects natural selection: harmful TEs are purged over time, but because selection acts with a time lag, young TEs still retain more harmful insertions than ancient ones. The efficiency of this purging depends on the strength of selection, which in turn varies with TE virulence as well as factors such as population size and genetic drift[62,63].

In line with the reduced harm of TEs with evolutionary age, selection on host defence was more effective in silencing younger than ancient TEs. Associations between TE methylation and both TE abundance and TE spreading efficiency were stronger in young than in ancient TEs (Supplementary Figs. 7 and Fig. 3c). This pattern reflects co-evolution between TE virulence and TE defence. As the virulence of older TEs declines, the selection pressure to silence ancient TEs weakens, while when young TEs invade, the pressure to silence them strengthens. Accordingly, many TE defence genes - especially from the piRNA pathway, which also mediates DNA methylation - were repeatedly under positive selection in termites (Supplementary Fig. 12).

## Variable arms race outcomes

Yet, not all ancient TEs are inactivated and harmless to the host. Some of the most active TEs, such as BovB and Tc1/mariner elements linked to ageing and senescence in *M. bellicosus*[31], appear to have ancient origins, as indicated by their high KD values (Fig. 4c). These TEs might have escaped silencing through methylation, making them currently evolutionary 'winners' of the arms race. Similarly, in *D. melanogaster*, TEs have been shown to employ counter-defence strategies against host silencing, including "anti-silencing" mechanisms[64]. We observed that the same TE families can be highly abundant and active in phylogenetically distant termite species, yet absent in others (Supplementary Data 1b). Such patterns may reflect recent invasions via horizontal transfer[65,66] and could elicit strong selection in the host, as has been shown in *Drosophila*[48]. Finally, some TEs can be co-opted by host genomes, contributing to regulatory rewiring, immune responses, or development[67–70], and thus provide raw material for novel functions and adaptation[71,72].

These variable outcomes reflect a pattern typical for interspecific associations known as symbioses, where gradients ranging from parasitism via commensalism to mutualism exist, with TE virulence and the nature of association between TEs and the host changing over time[1,14]. While it remains to be tested whether TEs can be beneficial mutualists in termites, it has been hypothesized that TE-induced gene tandem duplications might have been co-opted for caste-specific gene expression[33]. We expect that, on average, ancient TEs are more likely to develop such beneficial functions compared to younger ones due to long-term co-evolution with the host. Viewing TEs as evolving symbionts reconciles empirical findings and debates over whether they are harmful parasites, neutral genetic elements, or drivers of beneficial innovation. Ultimately, they are symbionts with effects on their hosts that vary over time and context.

## Methods

The study was conducted in accordance with the Nagoya protocol. The Parks and Wildlife Commission, Northern Territory and the Department of the Environment, Water, Heritage and the Arts provided permission to collect (permit number: 64452 and 71896) and export (permit number: PWS2019-AU-000897) termites from Australia. B. Sinsin provided collection permits for Benin and the Office Ivoirien des Parcs et Réserves provided sampling and export permits for samples from Côte d'Ivoire.

### Individual termite samples for genome sequencing

For *Cryptotermes secundus*, *Mastotermes darwiniensis*, *Reticulitermes grassei* and *Zootermopsis nevadensis*, a queen (for *R. grassei*) or a worker (for other species) had been collected from lab colonies housed at the University of Freiburg. *C. secundus* had been collected at Channel Island, Australia, in 2019; *M. darwiniensis* colony in Darwin, Australia, in 2022, *R. grassei* on Île d'Oléron, France, in 2020 (donated by Franck Dedeine), and *Z. nevadensis* in 2023 in California, USA. These colonies had been kept in climate chambers in Freiburg, Germany, and the individual termites were freshly killed in 2023 to extract DNA.

For the other species, samples had been collected in the field and stored in 100% ethanol or RNAlater (QIAGEN, Cat no. 76106) at −20 °C until DNA extraction: A *Trinervitermes geminatus* worker in Lamto, Côte d'Ivoire, in 2021; a *Odontotermes* sp.2 queen in the Comoé National Park, Côte d'Ivoire, in 2019; a *Macrotermes bellicosus* queen in the Comoé National Park, Côte d'Ivoire, in 2018, and a *M. bellicosus* minor soldier (female[31]) in the Pendjari National Park, Benin, in 2017. Besides, four major workers (males[31]) of *M. bellicosus* were collected from different colonies between 2018 and 2019 in the Comoé National Park, Côte d'Ivoire, to detect structure variants (SVs) in *M. bellicosus*.

### DNA extraction and sequencing

Because the body sizes of our sampled termite species differ by several orders of magnitude, to achieve individual genome sequencing, genomic DNA was extracted from heads (for *M. darwiniensis, Z. nevadensis* and *M. bellicosus* workers and soldier), dissected fat bodies (for *M. bellicosus* and *Odontotermes* sp.2 queens), or from gut-removed whole bodies (for the *R. grassei* queen, *T. geminatus* and *C. secundus* workers) (Supplementary Table 1). DNA extractions were carried out using the 2% cetyltrimethylammonium bromide (CTAB) protocol[73] with minor modifications (cut pipette tips to obtain high molecular weight DNA; digesting proteins with over-night proteinase K incubation to increase DNA yield). DNA sample quality was tested with a NanoDrop and DNA fragment sizes by agarose gel electrophoresis.

Depending on the amount of extracted DNA, low DNA input or standard DNA input libraries (Supplementary Table 1) were constructed for single-molecule real-time (SMRT) sequencing. For the four *M. bellicosus* minor workers used for detecting SVs, ultra-low DNA input libraries were constructed because of the small amount of DNA. Long high-fidelity (HiFi) reads were generated with the Pacific Biosciences (PacBio) Sequel II or the PacBio Revio system.

### Genome assembly

The genomes of the seven termite species were assembled with *Hifiasm* (v.0.19.5)[74] (with *--dual-scaf* for self-scaffolding and other default parameters for HiFi reads only genome assembly), a de novo assembler for PacBio HiFi reads. We noticed that the *T. geminatus* genome assembly had > 10% duplicated BUSCOs (Benchmarking Universal Single-Copy Orthologs) (v5.6.0 with insecta_odb10), significantly more than the other termite genome assemblies. We reason that the higher BUSCO duplication rate may be caused by the lower sequencing depth in *T. geminatus* because a lower sequencing depth can lead to false duplications in diploid genomes with high genomic heterozygosity[75]. To remove false duplications, we employed a second round of haplotig-duplication purging on the *T. geminatus* genome assembly with *purge_dups* (v.1.2.6) and with the *-e* option to only purge haplotypic duplications at the ends of contigs[76]. All genome assemblies were screened to remove potential contamination with Foreign Contamination Screening (FCS-GX)[77].

### Genome assembly evaluation

The genome assemblies of the seven termite species were evaluated using BUSCO (v.5.6.0 with insecta_odb10)[78] for gene content completeness, Merqury (v1.3)[79] for k-mer completeness ($k = 21$) and assembly consensus quality value (QV), and GenomeScope 2.0[80] for k-mer-based ($k = 21$) haploid genome length, heterozygosity, and repeat length estimation. For *T. geminatus*, both the pre- and post-deduplicated assemblies were assessed.

## Building a non-redundant termite repeat library

To improve the completeness of TE discovery and ensure comparability across species, we generated a non-redundant repeat library for termites. We first used RepeatModeler (v. 2.0.5)[81] (default parameters and -LTRStruct for running LTR structural search pipeline) to build eight de-novo repeat libraries from our seven termite genome assemblies (for *M. bellicosus*, we used the queen genome assembly) and the in-house *Cryptocercus punctulatus* (woodroach) genome. To reduce redundant TE families and to minimize false TEs due to high copy-number genes (such as olfactory receptor coding genes), we curated the repeat libraries using MCHelper (v1.6.6)[82] (default parameters and with -b option to filter out false TEs with BUSCO genes q[insecta_odb10]). We then merged the filtered repeat libraries into one redundant multi-species repeat library.

To reduce redundancy caused by orthologous TE families appearing multiple times in the redundant repeat library (e.g. when a TE family originated in the last common ancestor of *M. bellicosus* and *O.* sp2, resulting in two copies), we sorted the consensus sequences of TE families by size into the following categories: <500 bp, 500 bp–1 kb, 1 kb–2 kb, 2 kb–5 kb, 5 kb–10 kb, 10 kb–20 kb and > 20 kb. Within each size category, we clustered TE families based on sequence similarity ($\geq 90\%$) using the *ClusterPartialMatchingSubs.pl* script from RepeatModeler2, with the 90% cut-off chosen to balance sensitivity and specificity[83]. For each TE family cluster, we retained the longest sequence as the representative sequence. Finally, we obtained a non-redundant repeat library for termites by combining the representative sequences of the TE family clusters and that of the unique TE families, which only occurred in one species. TE sequences in the non-redundant repeat library were then classified with *RepeatModeler2*, using the Repbase (RepeatMasker Edition v20181026)[84] and the Dfam (v.3.8)[85] TE databases.

## Repeat annotation in genome assemblies

Using the classified non-redundant repeat library for termites, we annotated repeats in each genome assembly with *RepeatMasker* (v. 4.1.7)[86] (default parameters and using rmblast for searching sequences). To accurately quantify TE copy numbers, we applied One-code-to-find-them-all[87] (default parameters) to link TE fragments likely derived from the same TE copies. We then summarised the number of TE copies for each TE family in each genome assembly. To minimize false TE identifications, only TE families with more than 3 copies were retained for downstream analyses.

For superfamily-level analyses (e.g., PCA and dendrogram analyses), only classified TE families were included, as unclassified TEs cannot be assigned to superfamilies. For other analyses (e.g., TE age and the association between TE abundance and TE methylation at the family level), including or excluding unclassified TE families yielded similar results.

## Exons and introns annotation

Genomic regions of the seven termite species and the woodroach (*C. punctulatus*) were annotated by combining the Braker3 (v3.0.8)[88] and the StringTie (v2.2.1)[89] pipelines. For each species, the genome assembly was first soft-masked with RepeatMasker (see *Repeat annotation in genome assemblies*). For *M. darwiniensis*, *R. grassei*, *O.* sp. 2, and *T. geminatus*, in-house RNA-seq data (from another project) were used, following our in-house protocol for data generation and quality control[90]. In-house or published RNAseq data[31,34,90,91] were then mapped to the target genome with *Hisat2* (v. 2.2.1)[92] with the --dta option to report alignments tailored for transcript assemblers.

The masked genome was then annotated with both StringTie, which used only RNAseq data, and Braker3, which incorporated RNAseq data, protein data (Arthropoda orthologs from OrthoDB 11), and BUSCO data (insecta_odb10)[93]. The Braker and the StringTie annotations were then merged and annotated with Transdecoder (v5.71)[94]

(default parameters; BlastP [UniRef90] and Pfam searches enabled to retain ORFs based on homology), which predicts protein-coding genes and annotates exons, introns and untranslated regions.

Because the StringTie annotation can include transposons with coding regions, the merged genome annotations were scanned using InterProScan (v5.68)[95] (default parameters) so that genes with transposon-related domains (such as the transposase DDE domain) were removed from the final annotations.

For *M. bellicosus*, the queen genome assembly was selected as the representative genome for annotation, as it had higher quality than the soldier assembly, based on N50 and BUSCO metrics (Supplementary Table 1).

## Estimation of the evolutionary age of TE families

The ages of TE families were estimated based on their distribution across the termite phylogeny. In principle, if a TE family occurs in a specific lineage or species but is absent in others, we considered it to be lineage- or species-specific. To determine the presence or absence of TE families, we used effective copy numbers (ECN), calculated as the total sequence length of all TE copies in a family divided by its consensus sequence length. A TE family was considered present if its ECN > 5.

We classified TE families' ages into the following age groups:

*M. bellicosus*-specific: Present in *M. bellicosus* (either in the queen or soldier genome) but absent in other species. Since *M. darwiniensis* and *C. punctulatus* have been evolutionarily separated from other termite lineages for over 120 million years, any similar TEs found in both *M. bellicosus* and either of the two species are likely due to reintroduction. Therefore, we excluded *M. darwiniensis* and *C. punctulatus* from the TE age analysis in this and subsequent sections.

*O. sp2.*-specific: Present in *O. sp2.* but absent in other species.

Macrotermitinae-specific: Present in *M. bellicosus* and *O. sp2.* but absent in other species.
Termitidae-specific: Present in Macrotermitinae and *T. geminatus* but absent in other species.
Geoisoptera-specific: Present in Termitidae and *R. grassei* but absent in other species.

Ancient: Present in Geoisoptera and either *C. secundus* or *Z. nevadensis*. Since *C. secundus* and *Z. nevadensis* diverged from Geoisoptera between 120 to 140 million years ago, we classified TE families in these two species within the same age group.

Note that our analysis of TE age did not explicitly address the effects of horizontal transfer, which can make young TEs appear older once they enter different lineages. We defined TE age based on phylogenetic distribution. Any TE family that violated the termite phylogeny (e.g., present only in *C. secundus* and *M. bellicosus* but absent in other species), and was therefore likely derived from horizontal transfer was categorized as "unknown age" and excluded from our age-association analyses (e.g., TE age versus methylation level). Therefore, the impact of horizontal transfer on our conclusions should be limited.

## Quantification of TE abundance

TE abundance was quantified at the TE family and the TE superfamily level. For TE family level abundance quantification, we used the copy numbers derived from the summarized output of the One-code-to-find-them-all tool (see *Repeat annotation in genome assemblies*). For TE superfamily level quantification, we used the sum of the copy numbers of all TE families that have been classified under the same TE superfamily for each respective superfamily.

## Identification of structure variants with *M. bellicosus* population genomes

The Sniffles2 pipeline (v2.4)[96] (default parameters) was used for structure variants (SVs) identification. The PacBio reads of each *M. bellicosus* individual sample were mapped onto the *M. bellicosus* queen genome assembly using Minimap (v2.28)[97] (default parameters for HiFi reads). Population SVs were then identified with Sniffles multi-sample SV calling. For each SV location (which can include multiple SVs from different individuals), the longest SV sequence was extracted as the representative SV.

## Annotation and quantification of TE-associated SVs

We employed RepeatMasker to annotate SV sequences, using our termite non-redundant repeat library as the reference library. In addition, we also used *blastn* (with the *megablast* program) (v2.16)[98] to annotate the SV sequences by blasting against the repeat library and retaining the best-matched hits. We observed similar results for the annotated TE-associated SVs with both methods but the RepeatMasker method can also identify simple repeats. We therefore used the result from the RepeatMasker method for downstream analyses.

For the quantification of TE-associated SVs at the TE family level, we used the total number of SV copies from the same TE family. For the quantification at the superfamily level, we added the SV copies from all TE families that are classified under the same superfamily.

## Measurement of TE spreading efficiency

Because TE families with high abundances in the genome are more likely to be associated with SVs (Spearman's $r = 0.72$, $p < 1e\text{-}5$), we normalized the impact of TE abundance on the number of SVs using TE spreading efficiency. TE spreading efficiency was calculated as the total number of TE-associated SVs divided by the copy number of the target TE family within the genome (*M. bellicosus*).

## Inferring the evolutionary history of TE families based on Kimura Distances (KDs)

For each TE family in a genome, KDs were quantified for each TE copy by (1) aligning its sequence to the consensus sequence of the corresponding TE family, and (2) calculating the divergence using a modified Kimura 2-Parameter model, in which CG dinucleotide sites in the consensus sequence were treated with special consideration[51]. Following the quantification of the KDs, we inferred the evolutionary history of TE families by plotting their KD distribution as histograms. Within each histogram, the presence of a single mode with a low median KD ($< 10$) indicates a recent expansion of the TE family. Conversely, a single mode with a high median KD ($> 20$) indicates the presence of inactive TEs with ancient origins, often referred to as TE remnants. Additionally, a mixture of both, characterized by a bimodal distribution, implies a mixed evolutionary history.

## Identification of methylated DNA CpG sites

PacBio HiFi reads contain kinetic information that reflects CpG 5-Methylcytosine (5mC) modifications[99]. We thus used *Jasmine* (v. 2.0.0) (default parameters) to predict the CpG 5mC for our HiFi read sequences. The HiFi reads with annotated 5mC information were then mapped to the corresponding genome assembly with *minimap2 (pbmm2*, v. 1.12.0). To filter suspicious read alignments, we further used *samclip* to remove aligned reads with soft-clipped. With the aligned reads, we calculated the CpG-site methylation probabilities for each genome assembly using PacBio's *pb-CpG-tools* (*aligned_bam_to_cpg_scores* with the *model* pileup mode, v. 2.3.2), a machine-learning-based method for genomic CpG 5mC quantification.

For a robust determination of genomic CpG 5mC methylation, we only retained CpG sites with a read coverage between 10 to 50x; a CpG site was considered methylated when its CpG-site methylation probability was over 80%.

## Quantification of TE methylation levels

To quantify the CpG methylation level in different genomic regions and different TE families, we first used the R package *GenomicRanges*[100] to determine the total number of CpG sites in target genomic sequences. The percentage of methylated CpGs was then calculated as the number of methylated CpG sites divided by the total number of CpG sites in the target sequence. For methylation quantification at the TE family level (i.e., TE methylation), the percentage of methylated CpGs was calculated by summarising all CpGs sites across all copies of the target TE family. To avoid quantification uncertainty due to a small number of CpG sites, only TE families with over 30 CpG sites from all TE copies were retained for downstream analyses. For methylation quantification at the TE superfamily level, the percentage of methylated CpGs was calculated as the median of the percentages of methylated CpGs across all associated families.

Normalised TE methylation levels were then calculated as the ratio of the percentage of methylated CpGs in the target region or TE family/superfamily to the percentage of methylated CpGs in the genomic background. Therefore, a value $> 1$ indicates a higher percentage of CpG methylation relative to the genomic background, whereas a value $< 1$ signifies a lower percentage.

## Principal component analyses (PCA) and clustering analyses on TE abundance and methylation levels

For PCA and clustering analyses of TE abundance across the termite genome assemblies, TE abundances were summarized at the superfamily level (see *Quantification of TE abundance*). The superfamily level TE abundances were $\log_{10}$ transformed (with one added pseudo-count to avoid $\log_{10} 0$). For PCA and clustering analyses of TE methylation levels across the termite genome assemblies, TE methylation levels were summarized at the superfamily level (see *Quantification of CpG methylation levels in TEs*) and standardized within each genome assembly. The standardization within a genome assembly is necessary because the sequencing depth differences across genome assemblies can influence the confidence of modelling CpG methylation probability.

The transformed TE abundance and the standardized TE methylation levels were then analysed with the R package *FactoMineR*[101] and clustered with the R package *pheatmap*[102].

## Modelling the relationship between TEs and DNA methylation

The statistical relationship among TE family abundance, TE CpG methylation levels and TE age was tested using analysis of covariance (ANCOVA) in R[103] with the model:

$$\log 10(\text{TE abundance} + 1) \sim \text{TE methylation x TE age} \quad (1)$$

The statistical relationship among TE family spreading efficiency, TE methylation levels and TE age was tested using ANCOVA with the model:

$$\log 10(\text{TE spreading efficiency} + 1) \sim \text{TE methylation x TE age} \quad (2)$$

To examine the effect size of DNA methylation on TE spreading efficiency across different TE age groups, a Spearman's correlation test was performed to assess the relationship between TE methylation levels and TE spreading efficiency for each age group.

## Annotation of TE defence genes and homologous genes

We first searched FlyBase[104] and previous publications[31] to compile a set of TE defence-related genes, including those associated with DNA methylation, retrotransposon silencing, and piRNA processing. Next, we identified representative homologs of these TE defence genes in published termite/cockroach genomes with NCBI Orthologs[105]. For a TE defence gene with multiple homologs in the same species (e.g., due

to gene duplication), we retained all homologs and appended _N suffix to their names. For a TE defence gene with multiple single orthologs across different species, we retained the longest-matched ortholog as the representative.

We then collected the protein sequences of these representative TE defence gene orthologs as reference and aligned them to our termite and the woodroach genome assemblies with Miniprot (v0.13)[106]. For each target genome, we retained the best-matched Miniprot hits as the TE defence gene homologs.

For whole genome homolog annotation, we followed a similar procedure but used the protein sequences (longest protein isoforms) from *M. bellicosus* as the reference for Miniprot.

### Detection of positive selection in TE defence genes

To detect positive selection, we first aligned the orthologous coding nucleotide sequences from the termite and the woodroach genome assemblies. The alignment was done using the MACSE pipeline (v12.01) (default parameters), which employs codon-aware alignment and filters out non-homologous and mis-aligned fragments and sequences[107]. We then used HyPhy (Hypothesis testing using phylogenies) (v2.5.62)[108] to perform the branch-site unrestricted statistical test for episodic diversification (BUSTED) model[53] and the adaptive branch-site random effects likelihood (aBSREL) model[54]. The BUSTED model detected whether a target gene has experienced overall positive selection, while the aBSREL model identified the lineages or species in which the gene has undergone positive selection. To reduce false positive, false discovery rate (FDR) was applied on the p-values for multiple test correction.

### Reporting summary

Further information on research design is available in the Nature Portfolio Reporting Summary linked to this article.

## Data availability

The PacBio HiFi reads (including kinetic information used to infer methylation levels) for genome assembly and the *Macrotermes bellicosus* HiFi reads generated for population genomics have been deposited in the NCBI Sequence Read Archive (SRA) under BioProject accession PRJNA1033592. The termite genome assemblies have been deposited in the European Nucleotide Archive (ENA) under Study Accession number SRP470323. The non-redundant termite repeat library, repeat annotations, repeat abundance data, and summary tables of repeat methylation levels for all studied species are available on Zenodo (https://doi.org/10.5281/zenodo.17704238)[109] or GitHub (https://github.com/BitaoQiu/termites_TE_methylation)[110]. Source data are provided with this paper.

## Code availability

The shell and R scripts used to process the data are available on https://github.com/BitaoQiu/termites_TE_methylation.

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

## Acknowledgements

We thank Franck Dedeine for providing the *Reticulitermes grassei* lab colony for genome sequencing, Kiyoto Maekawa for access to the *Cryptocercus punctulatus* genome before it was published, and Daniela Schnaiter for carefully looking after the termite colonies. We acknowledge the support of the Baden-Württemberg High-Performance Computing facilities and the Deutsche Forschungsgemeinschaft (DFG) through grant INST 35/1597-1 FUGG. We gratefully acknowledge the data storage service SDS@hd supported by the Ministry of Science, Research and the Arts Baden-Württemberg and the DFG through grant INST 35/1803-1 FUGG and INST 35/1804-1 LAGG. Charles Darwin University (Australia), and especially S. Garnett and the Horticulture and Aquaculture team, provided logistic support to collect *Cryptotermes secundus* and *Mastotermes darwiniensis*. We thank N. Kone for logistic support in Côte d'Ivoire. This work was supported by the DFG Research Infrastructure West German Genome Center, project 407493903, as part of the Next Generation Sequencing Competence Network, project 423957469. Next Generation Sequencing was carried out at the West German Genome Center Düsseldorf. BQ was supported by a Humboldt Research Fellowship for Postdoctoral Researchers from the Alexander von Humboldt Foundation. In addition, this research was supported by the DFG with two grants to JK, one within the 'Sequencing cost in projects #1' initiative (DFG; KO1895/26-1, 29-1).

## Author contributions

B.Q. and J.K. designed the project and experiments. B.Q., D.E., J.K. collected samples. B.Q. and D.E. performed the experiment and collected

the data. B.Q. did the analyses. B.Q. and J.K. wrote the first draft of the manuscript. All authors contributed to the interpretation of experimental results and final manuscript revisions.

## Funding

## Competing interests
The authors declare no competing interests.
