## [Transparent Peer Review file · Nature Communications]

Arms races between selfish genetic elements and their host defence in termites

Corresponding Author: Dr Bitao Qiu

Version 0:

Reviewer comments:

Reviewer #1

(Remarks to the Author)

This manuscript describes an innovative and elegant study into the evolution of TEs, using termites as a model for their study. The authors sequenced several genomes from across the termite tree, and performed careful analyses of TEs, including their prevalence, location, age, and association with methylation. They show convincingly that the negative effects of TEs declines with their 'age', although some 'old' TEs still maintain activity. The manuscript is exceptionally well-organised and well-written, and I found it very interesting and readable. The topic is novel and is likely to be of interest to a general biological audience. It tackles a very interesting question related to genome evolution, and shows convincingly that TEs can be considered as a type of genetic symbiont.

I only have a few queries for the authors. Firstly, in Figure 1 and in other parts of the manuscript, is there a reason why information from the woodroach *C. punctulatus* is not included? These data would provide a very interesting comparison with the termite data. Secondly, did the authors correct for the use multiple tests in their positive selection? (methods, line 586). Because a very large number of tests are likely to have been performed, it is important to make sure that the appropriate p-value is used, in order to reduce the chance of false positives. Given these queries can be addressed, I would recommend publication in Nature Communications.

Minor grammatical suggestions.

Line 23: I think "evolutionarily..." is more appropriate as an adjective than "evolutionary" here.

Line 213: I suggest "...WAS negatively correlated WITH TE age".

Line 308: I suggest "selection was more effective in silencing younger than older TEs"

(Remarks on code availability)

Reviewer #2

(Remarks to the Author)

In this manuscript, Qiu and collaborators investigated the arms race between transposable elements activity and DNA methylation. Their hypothesis is grounded in the well-established concept that TEs are selfish. Therefore, the host must keep them under control, establishing an arms race between TEs and the host. The technologies and methods employed by the authors follow current gold-standard protocols, especially the use of long reads to analyze transposable elements. Among the hypothesis tested by the authors, they demonstrated that: 1) younger TEs have higher methylation compared to older TEs, which is expected due to their possible detrimental effects; 2) strong selective pressure for TE-derived SVs within exons; 3) ancient termite TEs have recent expansions in *M. bellicosus*; 4) Genes related to TE silence are under positive selection in the analyzed species. The study successfully and appealingly established a link between DNA methylation and the evolutionary dynamics of TEs.

Although some sentences are truncated or unfinished, the manuscript is well written. From my perspective, some conclusions require attention, so I recommend the manuscript as a major review. If these concerns are properly addressed, the manuscript can be suitable for publication in Nature Communications. Please, find below my point-by-point review.

Major review

L130-134: Although the authors found a significant p-value with a permutation test, and claim a consistent relationship with the findings from TE methylation and TE abundance, this result is not clear when comparing Figure 1C vs Figure 1E. For example, SINE/Alu was highlighted by the authors with high abundance in *Z. nevadensis* (Fig. 1C), but this TE is not shown on the TE methylation heatmap (Fig. 1E). DNA/Maverick has high abundance in Geoisoptera (Fig. 1C), but has systematically low methylation in all species, except *M. darwiniensis* (Fig. 1E). These are just two comparisons that don't support the authors' conclusions; there are more. I recommend that the authors represent the same TE superfamilies in 1C and 1E, and highlight the cases where the arms race hypothesis is observed, but also mention superfamilies that do not follow their expectation. I think this will provide a broader and more realistic interpretation of the data.

L188: TE-derived SVs do not necessarily correspond to "currently active TEs". They may correspond to inactive TEs that have been lost in one population and retained in another.

L208-L217: The authors should consider that young TEs are the ones likely to mobilize, which is not the expected for the majority of the old families. Please, if you agree, add this information to this paragraph.

L301-306: Could the authors extend their discussion regarding TEs inserted into exons? There is extensive literature showing beneficial insertions (domesticated, exapted insertions) in exons. In addition, could this observation of more young TEs in exons than old be associated with the fact that slightly deleterious young TEs will take longer to be purged by selection? While the observed remaining old insertions are only the ones neutral, since the slightly deleterious ones were removed. I agree with the overall conclusion of the authors, but I think these possibilities should be discussed.

Minor review

L37: Change "suborganismal" by "organism".

L38: Just cite 4,6, without "(e.g.)"

L38: Remove "good"

L40-53: There is a truncation in the text. I did my best to follow the rationale. Please, rewrite this paragraph.

L55: Remove "and references therein", and again, no need for "(e.g.)" before citations.

L56: Why do the authors consider piRNA and DNA methylation as the most important mechanisms over repressive chromatin marks, for example? I recommend the authors to include chromatin marks here.

L58: Remove "(for recent reviews, see)", just keep the citations.

L60-L61: By this sentence, the authors imply that TEs activity has been associated with ageing in termites, but none of the papers cited (27, 30) have evidence for that. Due to the misleading idea, I recommend removing "In termites, as well as in other organisms" by "In many organisms". I'd also strongly recommend adding "Although being a controversial topic", and then cite "Delanoue, R., Clot, C., Leray, C., Pihl, T., & Hudry, B. (2023). Y chromosome toxicity does not contribute to sex-specific differences in longevity. *Nature Ecology & Evolution*, 7(8), 1245-1256." TE activity and ageing was a solid idea in the past, but currently is a unsettled topic.

L63-64: I don't agree with this sentence "Because TEs generally make up about 50% of the genome size in termites 32,33, they present a large potential threat to the termite host.". From my perspective, a potential threat from TEs is given by their activity, not by the fraction they represent in the genome. For instance, TEs represent 50% of the human genome, and only 15-20% of the fruit fly. But in the fruit fly, they are way more active compared to humans. In the authors' words, they are a higher potential threat to genome stabilization in fruit flies than in humans.

L66: I am not sure if this can be a valid statement: "and which can permanently silence TEs". Methylation can change in many different ways: physiological changes and the environment for example. I'd change "permanently" to "can constitutively silence TEs". It gives the idea that methylation is established in homeostasis.

Figure 1A: In the genome assembly metrics, could the authors add the total of BUSCO genes covered? It's important to provide a metric of completeness, at least in terms of gene content.

Supplementary table 1 with sequencing data is missing.

L90: Please, clarify what is the metric of genome completeness represented by 98.2% and 100%.

L97: Remove "(Method)"

L98: Specify Figure 1A.

L100: The correlation doesn't support this conclusion. Please, change "is driven" to "might be driven".

L123: Truncated text.

L127-128: From my interpretation, PC1 separated Geoisoptera from *C. secundus*, while PC2 separated Geoisoptera from the other two blue termites.

Figure 2: Please, change grey scale to colors, it's easier to visualize.

Figure 3A caption: "... to the total number of insertions" per family? If so, please add this information.

L260-263: Long sentence - hard to follow. Please, rewrite it. Please, remove "and references therein)".

L281: truncated text or empty lines.

L323: truncated text.

L336-337: substitute "are" with "may be".

L400: Is there a reason to cluster TE families based on the 90% similarity cutoff? From Wicker et al., 2007, the cutoff would follow the 80-80 rule: 80% of similarity over 80% of the sequences.

L402: Is there any benefit of clustering all TEs into a single termite library, instead of using species-specific libraries to mask the genome? It seems the repeats will be overestimated in the genome because the diversity of repeats is inflated. I'm asking this as a minor review because I understand that reannotating TEs would mean redoing the whole paper. I just would like some clarification on why the authors decided to do it like that.

L431-L433: Super long sentence.

L568-595: Remove these lines.

(Remarks on code availability)

Reviewer #3

(Remarks to the Author)

Overall

This manuscript presents a valuable comparative analysis of transposable elements (TEs) and DNA methylation across six termite species. By integrating genome assemblies, TE annotations, and methylation data, the study offers novel insights into genome evolution and TE-host interactions in a eusocial insect clade. The incorporation of TE age distributions and methylation profiles adds an important layer of evolutionary context, and the manuscript addresses a timely and understudied question. However, several aspects of the presentation, interpretation, and methodological reporting could benefit from clarification or reorganization to improve the manuscript's impact and accessibility.

Abstract

No comments

Introduction

Lines 33, 43, 47, 49 (and throughout the paper): The manuscript frequently uses pronouns such as "they," "them," and "this," which sometimes lack clear antecedents. For example, in line 47, "they" could ambiguously refer to DNA transposons. To enhance clarity, consider replacing these with specific nouns where ambiguity may arise.

Line 42: The sentence reiterates the role of TEs as selfish genetic elements, already introduced in lines 39–40. To avoid redundancy, consider combining or rephrasing this sentence for smoother narrative flow.

Lines 61 and 63: Clarify the distinction between "old termites" and "old age kings or queens." It's unclear whether these refer to different biological categories or the same group. If they are distinct, further explanation would be helpful.

Line 60 – 66: The rationale for focusing on termites could be more clearly articulated. While the association between TE activity and aging is broadly relevant across taxa, it would strengthen the argument to explain what makes termites uniquely suited to address these questions, particularly in comparison to other eusocial insects with high TE loads. Additional context on caste longevity, reproduction, and methylation patterns would help justify the system choice.

Line 64: The focus on DNA methylation as the sole silencing mechanism is not fully explained. Since piRNA-mediated silencing was mentioned earlier, a brief justification for focusing exclusively on DNA methylation would help delineate the scope of the study.

Results

Line 84: In the introduction, the authors mentioned that TEs are highly active in old termites regardless of the cast. However, authors have sequenced a queen and a soldier from the same species to look at potential cast differences on TE abundance and DNA methylation. Since these two individuals share the same genome, in theory, there should not be any difference in TE abundance. However, if the age of the two specimens is different (old vs young) due to high TE activity in old specimens, there may be a difference in TE abundance. Therefore, I think it is important that the authors mention the age of the specimens being sequenced and further clarify in the introduction section the part about age and TE activity. Given that caste differences do not imply genomic differences, age-related variation in TE abundance should be clearly discussed. Please indicate the age of each specimen sequenced.

Line 90: BUSCO provides a useful but partial view of genome quality. To provide a more comprehensive assessment, consider reporting additional metrics such as k-mer completeness and assembly error rates (QV scores).

Line 90 and Supplementary table 1: It appears the BUSCO results reflect assemblies after haplotig removal. Please clarify whether BUSCO was run on pre- or post-deduplicated assemblies. For full transparency, it would be helpful to present

BUSCO scores from both versions where applicable.

Line 98: Consider rephrasing for smoother syntax:

Suggested revision: "Termite genome size correlated significantly with both the total size and the proportion of repetitive content..."

Line 105: The phrase "TE super/families" is ambiguous. Consider replacing with "TE superfamilies and families" or defining the term.

Line 107: Phylogeny of Figure 1. The methods section does not describe how the phylogeny (Figure 1A) was reconstructed. Please provide details on phylogenetic inference, including the data and methods used.

Line 136: A per-species summary of the relationship between TE methylation and abundance would clarify whether methylation consistently suppresses TE proliferation across all genomes.

Line 145: The manuscript does not address the possibility of horizontal transfer, a well-documented phenomenon in insects (e.g., Peccoud et al., 2017). Please clarify whether horizontal transfer was considered in the TE age analysis and how such events may influence results.

Line 173: TE age, as inferred from sequence divergence, does not necessarily reflect time of invasion. TEs may remain active long after initial insertion. To strengthen this section, the authors could analyze TE landscapes (TE abundance vs. divergence) to illustrate activity patterns over time. This would help distinguish recently active from genuinely young TEs.

Line 198: The observed depletion of TEs in exons is likely a result of purifying selection rather than insertion bias. Consider rephrasing to "retention bias" or "observed abundance bias" to avoid suggesting directional insertion preferences.

Line 206: Please elaborate on why lower exonic TE content in ancient TEs is interpreted as being less harmful. Is this pattern attributed to purifying selection that eliminates deleterious insertions over time, to reduced transpositional activity in older TEs, or to both? Additionally, the phrase "not active" raises the question of whether these TEs are being passively retained or gradually purged from the genome. Clarifying this point would strengthen the interpretation of the relationship between TE age and its potential harmfulness.

Discussion

Line 286: Consider including a scatter plot to visualize the relationship between genome size and TE content. Additionally, a breakdown of TE families contributing most to genome size variation could yield further insight.

Line 301: The distinction between TE age and TE activity could be made more explicit. Some TEs appear to be both ancient in origin and currently active, complicating interpretations of "young" vs. "old." If "young" refers to recent duplication or transposition rather than phylogenetic origin, please clarify this in the main text. Also, co-evolution between host silencing and TE activity should be interpreted with caution, as host silencing acts continuously regardless of TE origin. A clearer definition of "co-evolution" in this context would help ground the claim.

Methods

Lines 341 to 355: Please indicate the age or developmental stage of the queen and soldier specimens used for sequencing. Age-related differences in TE activity could influence interpretations.

Line 350: For clarity and reproducibility, include manufacturer details and catalog numbers for reagents (e.g., RNAlater, Invitrogen, cat. no. AM7020).

Line 375: Consider performing a k-mer-based genome profiling using tools such as Merqury, GenomeScope2, or FastK. These metrics (expected genome size, heterozygosity, duplication) would complement assembly statistics and strengthen the quality assessment.

Line 375 and onwards: For each computational tool used, please specify whether default parameters were used or indicate any custom settings.

Line 377: Report the version of BUSCO and the database lineage used. This is essential for reproducibility.

Line 413: Clarify how unknown or unclassified TEs were handled in your analysis.

Line 420: If the RNA-seq data used were newly generated, please include details on extraction, library preparation, sequencing platform, and data quality control.

Other comments

Consider briefly acknowledging other models (e.g., population structure, genetic drift, or epigenetic stochasticity) that could also influence TE dynamics.

Avoid long, compound sentences that list multiple findings. Splitting them into shorter sentences would enhance clarity and flow.

Ensure consistent use of terminology across the manuscript (e.g., “young,” “ancient,” “lineage-specific”).

Some genome assemblies appear to be fragmented, which may affect TE annotation. A brief discussion of this limitation and how intact TEs were identified would be appreciated.

(Remarks on code availability)

Version 1:

Reviewer comments:

Reviewer #1

(Remarks to the Author)

The authors have done an excellent job responding to my comments, and, as far as I can tell, the comments of the other reviewers. I am happy to recommend acceptance of the ms and congratulate the authors on a groundbreaking study.

(Remarks on code availability)

N/a

Reviewer #2

(Remarks to the Author)

The authors have addressed all my concerns, providing additional evidence to their results. I recommend the manuscript for publication in Nature Communications.

(Remarks on code availability)

Reviewer #3

(Remarks to the Author)

The authors have done a fantastic job in addressing the previous comments. Below is the only minor comment I have on the revised manuscript.

Line 94: The distinction between the total size of the repetitive content and the total proportion of the repetitive content is not quite clear. It appears they both represent the same information. Therefore, consider keeping only one.

(Remarks on code availability)

REVIEWER COMMENTS

Reviewer #1 (Remarks to the Author):

This manuscript describes an innovative and elegant study into the evolution of TEs, using termites as a model for their study. The authors sequenced several genomes from across the termite tree, and performed careful analyses of TEs, including their prevalence, location, age, and association with methylation. They show convincingly that the negative effects of TEs declines with their 'age', although some 'old' TEs still maintain activity. The manuscript is exceptionally well-organised and well-written, and I found it very interesting and readable. The topic is novel and is likely to be of interest to a general biological audience. It tackles a very interesting question related to genome evolution, and shows convincingly that TEs can be considered as a type of genetic symbiont.

We are glad that the reviewer liked our work and thankful for the constructive comments that helped to further improve our manuscript.

I only have a few queries for the authors. Firstly, in Figure 1 and in other parts of the manuscript, is there a reason why information from the woodroach *C. punctulatus* is not included? These data would provide a very interesting comparison with the termite data.

*The reviewer is correct that *Cryptocercus punctulatus* provides an interesting point of comparison with the termite data. However, the *C. punctulatus* genome was generated and kindly shared by Kiyoto Maekawa's group (<https://doi.org/10.1002/jez.b.23290>) and does not include genomic DNA methylation data. Therefore, we used the *C. punctulatus* genome only for the following purposes: (1) generating the non-redundant termite repeat library, (2) annotating the age of TE families, and (3) identifying genes under positive selection. We have also included TE superfamily abundance in *C. punctulatus* in **Supplementary Table 2**.*

*Principal component analysis and dendrogram analysis of TE superfamily abundance, including *C. punctulatus*, show that its TE superfamily distribution is distinct from that of termites (**Response Figure 1**). This supports our conclusion that TE composition exhibits strong phylogenetic signals. However, since our focus is on termites and genomic DNA methylation data for *C. punctulatus* are unavailable, we excluded it from the host–TE co-evolution analyses.*

*To provide a comparison with the termite data, we have now added genome size and repeat content information for *C. punctulatus* in a new **Supplementary Figure 1**, which illustrates the correlation between genome size and repeat content across termites and *C. punctulatus*.*

Response figure 1. TE superfamily abundance in termites and the woodroach. (A) PCA of TE superfamily abundances ($n = 71$) in the termite and the woodroach genomes, coloured according to species taxonomy. (B) Heatmap of TE superfamily abundance of the top 20 TE superfamilies with the highest abundance variation across species. TE superfamily abundances are shown on a log₁₀ scale.

Secondly, did the authors correct for the use multiple tests in their positive selection? (methods, line 586). Because a very large number of tests are likely to have been performed, it is important to make sure that the appropriate p-value

is used, in order to reduce the chance of false positives. Given these queries can be addressed, I would recommend publication in Nature Communications.

*We thank the Reviewer for pointing out this important aspect. While we had applied multiple testing correction when assessing positive selection for individual genes using the BUSTED model, we had not applied multiple testing correction across the full set of tests in the original version of the manuscript. **As suggested, we have now performed multiple testing correction using the false discovery rate (FDR) for both the TE defence genes and the genomic background in our positive selection analyses.** Although this reduced the number of TE defence genes identified as being under positive selection (from 22 to 15), the proportion of TE defence genes under positive selection (31%) remains significantly higher than in the genomic background (7%) (two-sided binomial test, $p < 1e-5$). **We have updated the Methods section, as well as the relevant results in the main text and the Supplementary Figure 12 accordingly.***

Minor grammatical suggestions.

Line 23: I think “evolutionarily...” is more appropriate as an adjective than “evolutionary” here.

Corrected.

Line 213: I suggest “...WAS negatively correlated WITH TE age”.

Corrected.

Line 308: I suggest “selection was more effective in silencing younger than older TEs.

Adjusted accordingly.

Reviewer #2 (Remarks to the Author):

In this manuscript, Qiu and collaborators investigated the arms race between transposable elements activity and DNA methylation. Their hypothesis is grounded in the well-established concept that TEs are selfish. Therefore, the host must keep them under control, establishing an arms race between TEs and the host. The technologies and methods employed by the authors follow current gold-standard protocols, especially the use of long reads to analyze transposable elements.

Among the hypothesis tested by the authors, they demonstrated that: 1) younger TEs have higher methylation compared to older TEs, which is expected due to their possible detrimental effects; 2) strong selective pressure for TE-derived SVs within exons; 3) ancient termite TEs have recent expansions in *M. bellicosus*; 4) Genes related to TE silencing are under positive selection in the analyzed species. The study successfully and appealingly established a link between DNA methylation and the evolutionary dynamics of TEs.

Although some sentences are truncated or unfinished, the manuscript is well written. From my perspective, some conclusions require attention, so I recommend the manuscript as a major review. If these concerns are properly addressed, the manuscript can be suitable for publication in Nature Communications.

Please, find below my point-by-point review.

We thank the reviewer for the positive words and thankful for the constructive comments that helped to further improve our work.

Major review

L130-134: Although the authors found a significant p-value with a permutation test, and claim a consistent relationship with the findings from TE methylation and TE abundance, this result is not clear when comparing Figure 1C vs Figure 1E. For example, SINE/Alu was highlighted by the authors with high abundance in *Z. nevadensis* (Fig. 1C), but this TE is not shown on the TE methylation heatmap (Fig. 1E). DNA/Maverick has high abundance in Geosiptera (Fig. 1C), but has systematically low methylation in all species, except *M. darwiniensis* (Fig. 1E). These are just two comparisons that don't support the authors' conclusions; there are more. I recommend that the authors represent the same TE superfamilies in 1C and 1E, and highlight the cases where the arms race hypothesis is observed, but also mention superfamilies that do not follow their expectation. I think this will provide a broader and more realistic interpretation of the data.

We thank Reviewer 2 for highlighting this important aspect. Due to space limitations, not all TE superfamilies were displayed in the heatmaps. In the original version, we showed only the top 20 TE superfamilies with the highest variation in abundance (Figure 1C) or in CpG 5mC levels across species (Figure 1E). As recommended, we have now updated Figures 1C and 1E to display the same set of TE superfamilies—specifically, the top 26 superfamilies that exhibit either high abundance variation or high methylation variation across the termite phylogeny.

*Regarding SINE/Alu, this TE superfamily is present only in *Z. nevadensis* and absent in other species (or present in only one or two copies, which we consider to be noise). As a result, DNA methylation information for SINE/Alu is available only for *Z. nevadensis*.*

*As for DNA/Maverick, the Reviewer is correct in noting that this superfamily shows high abundance and low methylation levels in Geosiptera, but low abundance and high methylation levels in both *Z. nevadensis* and *M. darwiniensis* (with the exception of *C. secundus*, which shows both low abundance and low methylation). This pattern suggests a recent expansion of DNA/Maverick elements in Geosiptera, which may not yet be effectively suppressed by host defenses.*

This interpretation is further supported by our analysis of Kimura distances (KD), as suggested by Reviewer 3. In Geosiptera, (1) the majority of high-abundance DNA/Maverick families exhibit low KD values, and (2) DNA/Maverick families with low KD values are hypomethylated (Response Figure 2). These findings indicate that these highly abundant DNA/Maverick families represent recent TE expansions and are not suppressed by host defence mechanisms—at least at the DNA methylation level.

*Notably, in the woodroach *Cryptocercus punctulatus*, DNA/Maverick families also exhibit low KD values, suggesting a convergent expansion of this superfamily in both Geosiptera and *Cryptocercus*. Although we do not have DNA methylation data for *C. punctulatus*,*

as the genome was generated and provided by another lab, we would predict that DNA/Maverick elements in this species are also hypomethylated.

Response Figure 2. Abundance, Kimura distance, and DNA methylation level of DNA/Maverick TE families. In Geosiptera, particularly in Termitidae: (A) DNA/Maverick TE families with high abundance (y-axis) tend to have low average Kimura distances (KDs) (x-axis), and (B) TE families with low KDs (x-axis) tend to have DNA methylation levels (y-axis) lower than the genomic background (dashed line). These patterns indicate recent expansions of DNA/Maverick TE families in Geosiptera that have not yet been effectively suppressed by host defences. A similar pattern is observed in the *Cryptocercus punctulatus* genome (although DNA methylation data are not available), suggesting convergent expansion of this TE superfamily.

While the relationship between TE abundance and DNA methylation at the superfamily level across the termite phylogeny provides useful insights at the general level—as demonstrated with the DNA/Maverick superfamily—we consider this level of resolution too coarse to fully capture the dynamics of TE–host co-evolution. This is because each TE superfamily encompasses multiple TE families with distinct evolutionary histories that can be species/lineage specific. As shown in other parts of our results (Figures 2–4), some TE families are young (lineage-specific) and active, others are ancient and dormant, and some are ancient but remain active. Moreover, TE families within the same superfamily can exhibit markedly different DNA methylation levels (Supplementary Figure 3) and each TE family can include both active and non-active TEs. Therefore, we believe it is important to first provide a general pattern at the superfamily level (e.g. to identify interesting superfamilies) but then to go into more detail and examine these patterns at the more informative TE family level or even at the individual TE copy level. This is what we have done in the manuscript and this why we have intentionally avoided overinterpreting trends at the superfamily level.

As suggested, we have now included additional interpretation at the superfamily level and added a new Supplementary Note 3 (under the heading Associations between TE methylation level and TE abundance at the superfamily level) to discuss the relationship between TE methylation and TE abundance at the superfamily level across the termite phylogeny.

L188: TE-derived SVs do not necessarily correspond to “currently active TEs”. They may correspond to inactive TEs that have been lost in one population and retained in another.

We thank the Reviewer for pointing this out. We agree that TE-derived structure variants (SVs) can include both active TEs and inactive TEs that have been lost in some populations.

However, by quantifying the distribution of SV insertions (compared to the reference genome) across the five individual termite samples (10 haplotypes), we found that most SV insertions occur in only one or two haplotypes (Response Figure 3). Because loss of a TE in the reference genome would lead to the same “insertion” being

detected in multiple haplotypes, the observation of haplotype-specific SV insertions suggests that these SVs are true TE insertions but not decayed TE sequences in the reference genome.

For SV deletions, it is more difficult to distinguish between true TE deletions (cut-and-paste events) and TE loss, as both scenarios can result in haplotype-specific SV deletions.

We have now replaced the term “reveal” with “hint at” to acknowledge the possibility that SVs may include some inactive TEs and added a new Supplementary Note 4 (under the heading *Inferring active TE families from TE derived structure variants*) to include this point.

Response Figure 3. Haplotype frequency distribution of SV deletion (left) and SV insertion (right) across the five individual termite samples comparing to the reference genome. The majority of SV insertions occur only in one or two haplotypes, indicating these SVs are derived from active TEs.

L208-L217: The authors should consider that young TEs are the ones likely to mobilize, which is not the expected for the majority of the old families. Please, if you agree, add this information to this paragraph.

We have now added this information and a related reference to the paragraph. While the general expectation is that young TEs are more likely to mobilize than older families, we note that there have been little systematic analyses examining the association between TE age and activity.

L301-306: Could the authors extend their discussion regarding TEs inserted into exons? There is extensive literature showing beneficial insertions (domesticated, exapted insertions) in exons. In addition, could this observation of more young TEs in exons than old be associated with the fact that slightly deleterious young TEs will take longer to be purged by selection? While the observed remaining old insertions are only the ones neutral, since the slightly deleterious ones were removed. I agree with the overall conclusion of the authors, but I think these possibilities should be discussed.

We thank the Reviewer for this constructive suggestion. We have expanded the discussion of exonic TE insertions and the interplay between natural selection and TE age (see L325–L331).

Minor review

L37: Change “suborganismal” by “organism”.

Changed.

L38: Just cite 4,6, without “(e.g.,)”

Changed and we have now removed all the “e.g.” before the citations.

L38: Remove “good”

Removed.

L40-53: There is a truncation in the text. I did my best to follow the rationale. Please, rewrite this paragraph.

We have now rewritten this paragraph.

L55: Remove “and references therein”, and again, no need for “(e.g.,” before citations.

Removed.

L56: Why do the authors consider piRNA and DNA methylation as the most important mechanisms over repressive chromatin marks, for example? I recommend the authors to include chromatin marks here.

The Reviewer is correct that chromatin marks is also a very important TE suppression mechanism. We have now added this in the introduction.

L58: Remove “(for recent reviews, see”, just keep the citations.

Removed.

L60-L61: By this sentence, the authors imply that TEs activity has been associated with ageing in termites, but none of the papers cited (27, 30) have evidence for that. Due to the misleading idea, I recommend removing “In termites, as well as in other organisms” by “In many organisms”.

We have replaced “In termites” with “In many organisms.” Please note that in Elsner et al. (2018) Longevity and transposon defense: the case of termite reproductives, we found that TE activity was up-regulated in the old major worker caste, but not in the queen or minor worker castes, indicating a caste-specific association between TE activity and aging in termites.

I'd also strongly recommend adding “Although being a controversial topic”, and then cite “Delanoue, R., Clot, C., Leray, C., Pihl, T., & Hudry, B. (2023). Y chromosome toxicity does not contribute to sex-specific differences in longevity. Nature Ecology & Evolution, 7(8), 1245-1256.” TE activity and ageing was a solid idea in the past, but currently is a unsettled topic.

We thank the Reviewer for noting that the role of TEs in aging remains an unsettled topic. We have now included this point along with the related reference.

L63-64: I don't agree with this sentence “Because TEs generally make up about 50% of the genome size in termites 32,33 , they present a large potential threat to the termite host.”. From my perspective, a potential threat from TEs is given by their activity, not by the fraction they represent in the genome. For instance, TEs represent 50% of the human genome, and only 15-20% of the fruit fly. But in the fruit fly, they are way more active compared to humans. In the authors' words, they are a higher potential threat to genome stabilization in fruit flies than in humans.

We thank the Reviewer for highlighting this logical flaw. We have revised the text accordingly to reflect this point (see L51 – L59).

L66: I am not sure if this can be a valid statement: “and which can permanently silence TEs”. Methylation can change in many different ways: physiological changes and the environment for example. I’d change “permanently” to “can constitutively silence TEs”. It gives the idea that methylation is established in homeostasis.

Thanks for this suggestion; we changed it accordingly.

Figure 1A: In the genome assembly metrics, could the authors add the total of BUSCO genes covered? It’s important to provide a metric of completeness, at least in terms of gene content.

*The information on the total BUSCO genes covered is available in **Supplementary Table1**. We have now also added this to **Figure 1A**.*

Supplementary table 1 with sequencing data is missing.

*Information on sequencing data is in **Supplementary Table1** under the heading **PacBio data statistics**. The raw PacBio sequencing data has been uploaded to NCBI under the project number PRJNA1033592 (<https://dataview.ncbi.nlm.nih.gov/object/PRJNA1033592?reviewer=14aemg42gu201rtpms2o2e1huh>).*

L90: Please, clarify what is the metric of genome completeness represented by 98.2% and 100%.

It’s the total BUSCO genes covered. We have now added a clarification.

L97: Remove “(Method)”

Removed.

L98: Specify Figure 1A.

Specified.

L100: The correlation doesn’t support this conclusion. Please, change “is driven” to “might be driven”.

Respectfully, we believe the sentence is correct as written. “Is driven” refers to “the hypothesis” that genome size variation is driven by TE expansion. The fact that we only observe a correlation, and not causal evidence, is already reflected in our use of “supports” rather than “confirms.”

L123: Truncated text.

Rephased.

L127-128: From my interpretation, PC1 separated Geoisoptera from C. secundus, while PC2 separated Geoisoptera from the other two blue termites.

We thank the Reviewer for pointing out this. We have now changed the interpretation.

Figure 2: Please, change grey scale to colors, it’s easier to visualize.

*We have now employed colour scale in **Figure 2A** and **Figure 4A**.*

Figure 3A caption: "... to the total number of insertions" per family? If so, please add this information.

Added.

L260-263: Long sentence - hard to follow. Please, rewrite it. Please, remove "and references therein").

Rewritten.

L281: truncated text or empty lines.

Rephased.

L323: truncated text.

Rephased.

L336-337: substitute "are" with "may be".

We think that "are" is the correct term. TEs are symbionts, as they live in close association with the host. The term "symbiont" is neutral and does not imply whether the relationship is parasitic, mutualistic, or commensalistic.

L400: Is there a reason to cluster TE families based on the 90% similarity cutoff? From Wicker et al., 2007, the cutoff would follow the 80-80 rule: 80% of similarity over 80% of the sequences.

*We employed a 90% similarity cut-off to balance sensitivity and specificity. This follows the recommendation from [<https://doi.org/10.1371/journal.pone.0016526>] and is supported by our empirical observations, as the 80-80-80 rule can fail to capture species-specific TE families when clustering TE families across species. Moreover, the 80-80-80 rule was originally developed to group TE copies within the same family, whereas our goal is to cluster consensus sequences from different TE families across different species to reduce redundancy. Accordingly, we applied a more stringent cut-off. **This rationale has now been added to the Methods section.***

L402: Is there any benefit of clustering all TEs into a single termite library, instead of using species-specific libraries to mask the genome? It seems the repeats will be overestimated in the genome because the diversity of repeats is inflated. I'm asking this as a minor review because I understand that reannotating TEs would mean redoing the whole paper. I just would like some clarification on why the authors decided to do it like that.

We employed a clustering approach (first annotating TEs in each genome separately, and then clustering them into a single non-redundant library) for two reasons:

- 1. **Improved completeness of TE discovery.** TE annotation tools such as RepeatModeler use a sampling strategy, which may result in missing TE families or fragmented consensus sequences when applied to a single genome. By annotating all species individually and then clustering the results, we can recover a broader set of TE families and generate more complete, full-length consensus sequences.*
- 2. **Comparability across species.** By clustering all TE families into a unified termite TE library, we ensure that the same reference library is used to annotate all genomes. This enables us to identify homologous TEs across species and, crucially, to estimate the age of TE families based on species- or lineage-specificity—an essential step for reconstructing the evolutionary history of TEs across the termite phylogeny.*

We have now added the rationale behind employing a single non-redundant library in the Method section.

L431-L433: Super long sentence.

Rephased.

L568-595: Remove these lines.

*We are unsure why we should remove this part on **Annotation of TE defence genes and homologous genes**. We think it is essential for a reader to understand what we did and therefore, we respectfully did not remove this section.*

Reviewer #3 (Remarks to the Author):

Overall

This manuscript presents a valuable comparative analysis of transposable elements (TEs) and DNA methylation across six termite species. By integrating genome assemblies, TE annotations, and methylation data, the study offers novel insights into genome evolution and TE-host interactions in a eusocial insect clade. The incorporation of TE age distributions and methylation profiles adds an important layer of evolutionary context, and the manuscript addresses a timely and understudied question. However, several aspects of the presentation, interpretation, and methodological reporting could benefit from clarification or reorganization to improve the manuscript's impact and accessibility.

We thank the reviewer for the helpful comments that further improved our manuscript

Abstract

No comments

Introduction

Lines 33, 43, 47, 49 (and throughout the paper): The manuscript frequently uses pronouns such as “they,” “them,” and “this,” which sometimes lack clear antecedents. For example, in line 47, “they” could ambiguously refer to DNA transposons. To enhance clarity, consider replacing these with specific nouns where ambiguity may arise.

We thank the reviewer for pointing this out. We have now checked the manuscript systematically and replaced pronouns with specific nouns where ambiguity may arise.

Line 42: The sentence reiterates the role of TEs as selfish genetic elements, already introduced in lines 39–40. To avoid redundancy, consider combining or rephrasing this sentence for smoother narrative flow.

We have now rephased this sentence to reduce redundancy.

Lines 61 and 63: Clarify the distinction between “old termites” and “old age kings or queens.” It’s unclear whether these refer to different biological categories or the same group. If they are distinct, further explanation would be helpful.

We apologize for the confusion. We have rewritten the section and now use “age” only in the sense of individuals being physiologically old (i.e. senescent), which we indicate explicitly by using the term “senescent.”

Line 60 – 66: The rationale for focusing on termites could be more clearly articulated. While the association between TE activity and aging is broadly relevant across taxa, it would strengthen the argument to explain what makes termites uniquely suited to address these questions, particularly in comparison to other eusocial insects with high TE loads. Additional context on caste longevity, reproduction, and methylation patterns would help justify the system choice.

*We thank the Reviewer for this constructive suggestion. In brief, compared to other social insects such as ants and bees, termite genomes contain higher proportions of TEs and exhibit higher levels of DNA methylation. Notably, DNA methylation is largely absent in holometabolous insects, including flies and honeybees. **We have now revised the paragraph to better justify why termites provide a suitable system to study the arms race between TEs and DNA methylation (see L51 – L59).***

Line 64: The focus on DNA methylation as the sole silencing mechanism is not fully explained. Since piRNA-mediated silencing was mentioned earlier, a brief justification for focusing exclusively on DNA methylation would help delineate the scope of the study.

*We thank the Reviewer for this constructive suggestion. While piRNA plays an important role in TE defence, DNA methylation is more widespread in termites than in most other insects, making termites a particularly interesting system for studying DNA methylation. **We have now included this point in the introduction.***

Results

Line 84: In the introduction, the authors mentioned that TEs are highly active in old termites regardless of the cast. However, authors have sequenced a queen and a soldier from the same species to look at potential cast differences on TE abundance and DNA methylation. Since these two individuals share the same genome, in theory, there should not be any difference in TE abundance. However, if the age of the two specimens is different (old vs young) due to high TE activity in old specimens, there may be a difference in TE abundance. Therefore, I think it is important that the authors mention the age of the specimens being sequenced and further clarify in the introduction section the part about age and TE activity. Given that caste differences do not imply genomic differences, age-related variation in TE abundance should be clearly discussed. Please indicate the age of each specimen sequenced.

*We thank the Reviewer for this suggestion. Chronologically, the queen sample was older than the soldier sample (>7 years vs. a few months), but absolute lifespan differs between castes and species. Considering relative age (i.e. the age of the sequenced individuals in relation to the typical maximum lifespan of their caste/species), all individuals used were of intermediate age. Therefore, we do not expect age to have affected our data. **We have now provided the available age information for our genome samples in Supplementary Table 1.***

Line 90: BUSCO provides a useful but partial view of genome quality. To provide a more comprehensive assessment, consider reporting additional metrics such as k-mer completeness and assembly error rates (QV scores).

*We thank the Reviewer for the suggestion. **We have now included k-mer completeness, assembly consensus quality values (QV) and assembly error rates for our genome assemblies (for *T. geminatus*, we provide both values before and after haplotig removal) in the Supplementary Table 1.** We would like to note that while k-mer completeness is useful, it may underestimate the actual quality of the genome assemblies, as the raw reads can contain sequence contamination (e.g., from endosymbiont genomes), whereas the final assemblies have been cleaned from potential contamination using the NCBI Foreign Contamination Screening pipeline.*

Line 90 and Supplementary table 1: It appears the BUSCO results reflect assemblies after haplotig removal. Please clarify whether BUSCO was run on pre- or post-deduplicated assemblies. For full transparency, it would be helpful to present BUSCO scores from both versions where applicable.

*The BUSCO results reflect the assemblies that were used for the TE and DNA methylation analyses. As described in the Methods section, only the *T. geminatus* genome was processed with haplotig removal, as this genome showed a higher number of duplicated BUSCOs. **We have now clarified this in the Method section and provided the BUSCO results for both the pre- and post-deduplicated *T. geminatus* assemblies in Supplementary Table 1.***

Line 98: Consider rephrasing for smoother syntax:

Suggested revision: "Termite genome size correlated significantly with both the total size and the proportion of repetitive content..."

Rephased.

Line 105: The phrase "TE super/families" is ambiguous. Consider replacing with "TE superfamilies and families" or defining the term.

We now replaced it with "TE superfamilies". The PCA and the dendrogram analyses compared TE abundance across species at the superfamily level.

Line 107: Phylogeny of Figure 1. The methods section does not describe how the phylogeny (Figure 1A) was reconstructed. Please provide details on phylogenetic inference, including the data and methods used.

*We apologize for the confusion. The termite phylogeny was based on Hellemans et al. (Nature Communications, 2024; <https://doi.org/10.1038/s41467-024-51028-y>) and Buček et al. (Current Biology, 2019; <https://doi.org/10.1016/j.cub.2019.08.076>). While these references were cited in the Figure 1 legend, they were not included in the main text. **We have now added the citations to the Results section.***

Line 136: A per-species summary of the relationship between TE methylation and abundance would clarify whether methylation consistently suppresses TE proliferation across all genomes.

*A per-species summary of the relationship between TE methylation and abundance has been provided in **Supplementary Figure 8**. TE methylation and abundance were negatively correlated in all species (Spearman's correlation test; all $p < 1e-5$, except for *C. secundus*). **We have now added this information to the legend of Supplementary Figure 8.***

Please note that while the association between TE methylation and abundance can be informative for assessing whether DNA methylation suppresses TEs, TE abundance at the genomic level reflects both inactive and active TEs. Therefore, it is more meaningful to examine the association between TE methylation and TE activity (as quantified by structural variants at the population level), which we have analyzed in the later part of our manuscript.

Line 145: The manuscript does not address the possibility of horizontal transfer, a well-documented phenomenon in insects (e.g., Peccoud et al., 2017). Please clarify whether horizontal transfer was considered in the TE age analysis and how such events may influence results.

We thank the Reviewer for the suggestion. Horizontal transfer of TEs can indeed make young TEs appear older, as it results in the presence of the same young TE families across different lineages. While we had not previously discussed this effect, we did notice some TE families that were highly abundant and with low KD values across phylogenetically distant species.

*Although we did not explicitly model horizontal transfer in our TE age analysis, TE age was defined based on phylogenetic distribution. Any TE family that violated the termite phylogeny (e.g., only present in *C. secundus* and *M. bellicosus* but absent in other species), and thus likely derived from horizontal transfer, was categorized as of "unknown age" and excluded from our age-association analyses (e.g., TE age versus methylation level). Therefore, horizontal transfer should have only a limited impact on our main conclusions. **We have now added a discussion of this issue in the Method section.***

Line 173: TE age, as inferred from sequence divergence, does not necessarily reflect time of invasion. TEs may remain active long after initial insertion. To strengthen this section, the authors could analyze TE landscapes (TE abundance vs. divergence) to illustrate activity patterns over time. This would help distinguish recently active from genuinely young TEs.

*We thank the Reviewer for this constructive suggestion. Unlike many previous studies that inferred TE age based on TE landscapes (distribution of Kimura's distance, KD), we inferred the age of TE families based on their distribution across the termite phylogeny. We believe this approach provides higher resolution for the timing of TE invasions compared to KD values alone, which mainly reflect recent expansion and can assign the same KD value to TE families of different evolutionary ages. Analysis of KD values across TE age groups supports this point: while KD values of ancient TE families are much higher than those of younger families, as expected because most ancient TEs are remnants, the KD values of Macrotermitinae-specific TE families are only slightly higher than those of *M. bellicosus*-specific TE families (**Response Figure 4**).*

Interestingly, by integrating KD values with TE phylogenetic distribution, we found that while TE methylation levels generally declined with TE age regardless of recent expansion, the more diverged (most likely non-active) TE families

($KD > 20$) had significantly higher methylation levels than the recent-expanded (likely still active) TE families ($KD < 20$) across all TE age groups, except for ancient TE families (**Response Figure 5**). This result is consistent with the expectation that DNA methylation suppresses TE proliferation and that methylation is no longer maintained by natural selection in ancient non-active TE families that no longer pose a threat to the host. **We have now added this extra analysis in the Result section (see L174 – L185).**

Response Figure 4. Boxplots of KD values of TE families across different TE age groups. The mean KD values of ancient TE families were significantly higher than those of younger families (two-sided t-test; $p < 1e-5$), as expected since most ancient TE families are TE remnants. However, there was no significant difference between the KD values of *M. bellicosus*-specific and *Macrotermitinae*-specific TE families ($p = 0.33$), indicating that KD values have limited resolution for inferring the age of younger TE families.

Response Figure 5. Boxplots of TE methylation levels across different TE age groups. Within each age group, TE families are stratified into the less diverged (indicating recent expansion and likely still active) (Kimura's distance, $KD < 20$; red) and the more diverged (most likely non-active) ($KD > 20$, blue; $KD > 30$, black) categories. Overall, TE methylation levels declined with increasing TE age. However, the more diverged TE families exhibited significantly higher methylation levels than the recently expanded families in all age groups, except for the most ancient TEs.

Line 198: The observed depletion of TEs in exons is likely a result of purifying selection rather than insertion bias. Consider rephrasing to “retention bias” or “observed abundance bias” to avoid suggesting directional insertion preferences.

We thank the Reviewer for the suggestion. We have now replaced ‘insertion bias’ with ‘observed abundance bias’.

Line 206: Please elaborate on why lower exonic TE content in ancient TEs is interpreted as being less harmful. Is this pattern attributed to purifying selection that eliminates deleterious insertions over time, to reduced transpositional activity in older TEs, or to both? Additionally, the phrase “not active” raises the question of whether these TEs are being passively retained or gradually purged from the genome. Clarifying this point would strengthen the interpretation of the relationship between TE age and its potential harmfulness.

*We thank the Reviewer for this constructive feedback. We acknowledge that our explanation was not sufficiently clear. We agree that the lower exonic TE content in ancient compared to young TEs is largely the result of purifying selection, although other factors may also contribute. However, we think reduced transpositional activity should not systematically affect exonic insertion bias. **We have now elaborated on this point in the Discussion section Co-evolution of virulence and host defence. In this revised section, we also explicitly define co-evolution and outline the evidence we found for it, highlighting the role of natural selection as well as other contributing factors (see L313 – L331).***

Discussion

Line 286: Consider including a scatter plot to visualize the relationship between genome size and TE content. Additionally, a breakdown of TE families contributing most to genome size variation could yield further insight.

We have added a new Supplementary Figure 1 to visualize the relationship between genome size and TE content across termites and the woodroach. We further sorted TE superfamilies by their size variation among termite genomes and found that RTE-BovB (LINE), tRNA (SINE), Gypsy (LTR retrotransposon), TcMar-Mariner (DNA transposon), L2 (LINE), and Tc1/mariner (DNA transposon) show the highest variation. Notably, BovB and Tc1/mariner are also among the top three active TE superfamilies in *M. bellicosus* (together with Ginger-2, a DNA transposon that also exhibits high size variation). These findings suggest that genome size variation among termites can be partly attributed to TE families that remain active. **We have now included this point in a new Supplementary Note 5 under the heading Contribution of TE superfamilies to genome size variation among termites.**

Line 301: The distinction between TE age and TE activity could be made more explicit. Some TEs appear to be both ancient in origin and currently active, complicating interpretations of “young” vs. “old.” If “young” refers to recent duplication or transposition rather than phylogenetic origin, please clarify this in the main text. Also, co-evolution between host silencing and TE activity should be interpreted with caution, as host silencing acts continuously regardless of TE origin. A clearer definition of “co-evolution” in this context would help ground the claim.

*We thank the Reviewer for this suggestion. We recognize that our terminology was complicated and potentially misleading. **We have therefore re-checked and adjusted it.** In this manuscript, the definitions of “young” and “old” TE families are based on their phylogenetic origins rather than on whether they are currently active. We now consistently use the terms ‘young’ and ‘ancient’ (or ‘associations with age’) when referring to evolutionary age differences between TE families. Therefore, ‘ancient’ does not always imply that TEs are inactive (and vice versa). The actual activity of TEs was inferred from the SV analyses, and we refer to TEs as active only in this context. In addition, by measuring Kimura distance (KD) of TE families, we identified recently expanded TE families or TE copies that are likely still active. **In this situation, we now consistently use the terms ‘recently expanded’ and ‘more diverged’ TEs.***

*With regard to ‘co-evolution’, we agree with the Reviewer that host silencing applies continuously regardless of TE age. **We have therefore added a definition of co-evolution (see L314 – L315) as the reciprocal effects that TEs and their hosts exert on each other through natural selection.***

Methods

Lines 341 to 355: Please indicate the age or developmental stage of the queen and soldier specimens used for sequencing. Age-related differences in TE activity could influence interpretations.

We have added this information to Supplementary Table 1. Although there was considerable variation in chronological age among the individuals used, all were of intermediate age relative to the typical lifespan of their species and caste.

Line 350: For clarity and reproducibility, include manufacturer details and catalog numbers for reagents (e.g., RNAlater, Invitrogen, cat. no. AM7020).

We have now added this information.

Line 375: Consider performing a k-mer-based genome profiling using tools such as Merqury, GenomeScope2, or FastK. These metrics (expected genome size, heterozygosity, duplication) would complement assembly statistics and strengthen the quality assessment.

We have now added k-mer based quality assessment results from Merqury and GenomeScope2 to complement the assembly statistics (see Supplementary Table 1).

Line 375 and onwards: For each computational tool used, please specify whether default parameters were used or indicate any custom settings.

We have systematically reviewed the Methods section and clarified in each case whether default parameters were used or if custom settings were applied.

Line 377: Report the version of BUSCO and the database lineage used. This is essential for reproducibility.

We have now reported the version of BUSCO and the database lineage used.

Line 413: Clarify how unknown or unclassified TEs were handled in your analysis.

*For superfamily-level analyses (e.g., PCA and dendrogram analyses), only classified TE families were included, as unclassified TEs cannot be assigned to superfamilies. For other analyses (e.g., TE age and the association between TE abundance and TE methylation at the family level), including or excluding unclassified TE families produced similar results. Because unclassified TEs may contain false positives, we reported only the results with classified TEs. **We have now made this clear in the method section.***

Line 420: If the RNA-seq data used were newly generated, please include details on extraction, library preparation, sequencing platform, and data quality control.

*For *Macrotermes bellicosus*, *Cryptotermes secundus*, and *Zootermopsis nevadensis*, published RNA-seq data were used for exon and intron annotation. For *Mastotermes darwiniensis*, *Reticulitermes grassei*, *Odontotermes* sp. 2, and *Trinervitermes geminatus*, unpublished RNA-seq data from other projects were used. Data generation and quality control followed our in-house protocol (see [DOI: 10.1038/s42003-021-01892-x](https://doi.org/10.1038/s42003-021-01892-x)). **This information has now been included in the Method section.***

Other comments

Consider briefly acknowledging other models (e.g., population structure, genetic drift, or epigenetic stochasticity) that could also influence TE dynamics.

Thanks for pointing this out. We briefly acknowledge the effect of other factors on TE dynamics in the discussion in the section: Co-evolution of virulence and host defence (see L329 – L331)

Avoid long, compound sentences that list multiple findings. Splitting them into shorter sentences would enhance clarity and flow.

We have now systematically checked the manuscript to reduce long compound sentences.

Ensure consistent use of terminology across the manuscript (e.g., “young,” “ancient,” “lineage-specific”).

We thank the Reviewer for this suggestion. We have now systematically checked the manuscript to ensure consistent use of terminology.

Some genome assemblies appear to be fragmented, which may affect TE annotation. A brief discussion of this limitation and how intact TEs were identified would be appreciated.

*We thank the Reviewer for this suggestion. We agree that genome assembly continuity can influence TE annotation and quantification, for example by reducing the recovery of full-length TE consensus sequences or inflating TE abundance estimates in fragmented assemblies. However, we observed little impact in our analyses. For instance, although the queen and soldier *M. bellicosus* assemblies differ greatly in contiguity (364 vs. 6,173 contigs; 15,380,714 bp vs. 528,038 bp for N50), they show nearly identical repeat contents (55.83% vs. 55.58%) and highly similar TE superfamily abundances compared to other species (**Figure 1B and C**). This robustness may reflect our use of a non-redundant repeat library, which improves recovery of full-length TE consensus sequences, and/or the fact that phylogenetic signals in TE abundance are stronger than potential biases from assembly continuity, especially since all assemblies were generated from long-read data.*